# SPORTU: A Comprehensive Sports Understanding Benchmark for Multimodal Large Language Models

**Haotian Xia**[1,2*] **Zhengbang Yang**[3] **Junbo Zou**[2] **Rhys Tracy**[4] **Yuqing Wang**[5]
**Chi Lu**[2] **Christopher Lai**[4] **Yanjun He**[2] **Xun Shao**[2] **Zhuoqing Xie**[4]
**Yuan-fang Wang**[4] **Weining Shen**[2] **Hanjie Chen**[1]
[1]Rice University [2]University of California, Irvine [3]George Mason University
[4]University of California, Santa Barbara [5]Stanford University
{xiah6, weinings}@uci.edu, hanjie@rice.edu

## Abstract

Multimodal Large Language Models (MLLMs) are advancing the ability to reason about complex sports scenarios by integrating textual and visual information. To comprehensively evaluate their capabilities, we introduce SPORTU, a benchmark designed to assess MLLMs across multi-level sports reasoning tasks. SPORTU comprises two key components: SPORTU-text, featuring 900 multiple-choice questions with human-annotated explanations for rule comprehension and strategy understanding. This component focuses on testing models' ability to reason about sports solely through question-answering (QA), without requiring visual inputs; SPORTU-video, consisting of 1,701 slow-motion video clips across 7 different sports and 12,048 QA pairs, designed to assess multi-level reasoning, from simple sports recognition to complex tasks like foul detection and rule application. We evaluated four prevalent LLMs mainly utilizing few-shot learning paradigms supplemented by chain-of-thought (CoT) prompting on the SPORTU-text part. GPT-4o achieves the highest accuracy of 71%, but still falls short of human-level performance, highlighting room for improvement in rule comprehension and reasoning. The evaluation for the SPORTU-video part includes 6 proprietary and 8 open-source MLLMs. Experiments show that models fall short on hard tasks that require deep reasoning and rule-based understanding. GPT-4o performs the best with only 57.8% accuracy on the hard task, showing large room for improvement. We hope that SPORTU will serve as a critical step toward evaluating models' capabilities in sports understanding and reasoning. The dataset is available at `https://github.com/chili-lab/SPORTU`.

## 1 Introduction

The sports domain has witnessed a surge in interdisciplinary research, combining Natural Language Processing (NLP) and computer vision (CV) to tackle a wide range of applications. For instance, NLP-based approaches have been leveraged for automated sports news generation, producing detailed summaries and news articles from game data (Huang et al., 2020a; Wang et al., 2022b). Concurrently, hate speech detection has been employed to mitigate the impact of toxic content on social media (Vujičić Stanković & Mladenović, 2023), enabling athletes to maintain focus on their game. In the realm of CV, action recognition (Zhu et al., 2022b; Li et al., 2021), player detection (Maglo et al., 2022; Vandeghen et al., 2022), and tactical analysis (He et al., 2024b; Xia et al., 2023) have been explored, enhancing visual content for analysis and fan engagement. The recent emergence of Large Language Models (LLMs) (OpenAI, 2024c; AI@Meta, 2024; Jiang et al., 2024; Anil et al., 2023) and Multimodal LLMs (MLLMs) (OpenAI, 2024b; Gemini Team, 2024a; Anthropic, 2024a; Lin et al., 2023) has further accelerated this trend, enabling researchers to develop novel tasks such as AI-assisted refereeing (Held et al., 2024a; 2023), where models analyze game videos to identify

---

*Work done at Rice University

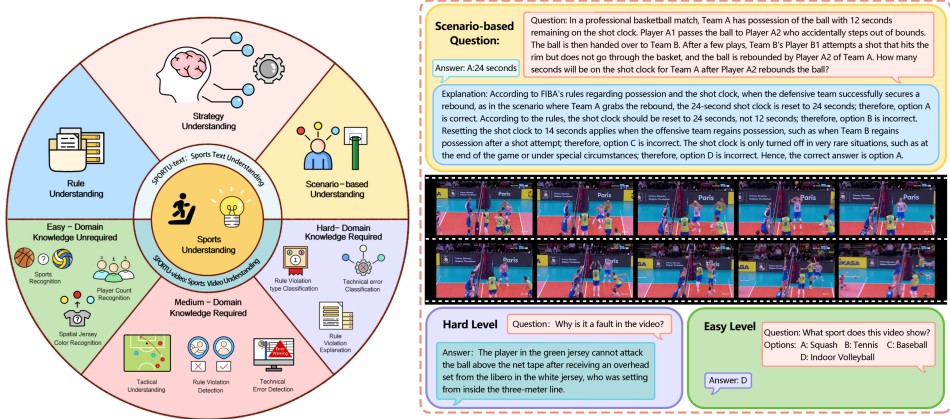

Figure 1: SPORTU consists of two parts: SPORTU-text, which evaluates sports understanding through text-based multiple-choice questions, and SPORTU-video, which assesses multi-level reasoning through video-based QA tasks. SPORTU provides a comprehensive sports understanding evaluation of multi-leveled reasoning beyond perception. Right side is a sample from the scenario-based question in SPORTU-text, along with examples of both hard-level and easy-level questions from SPORTU-video.

fouls and violations, and interactive sports education (Zhang et al., 2025; Zeng et al., 2023), where users engage with LLMs to learn rules, strategies, and game-related content.

However, the effectiveness of these applications depends crucially on the model's deep understanding of sports knowledge. While LLMs act as study guides, helping users learn rules and general strategies through text, MLLMs extend this knowledge to video-based tasks that require video and action perception, as well as the ability to connect movements with context-based rules. For example, when answering a question like "Why is it a rule violation in the video?", a model must distinguish the series of actions performed by the players and understand the corresponding rules that define the fault. This capability underscores the deeper reasoning required for real-world sports comprehension. The challenges, ranging from general video recognition to deep sports knowledge reasoning, highlight the need for a dedicated sports-focused question-answering (QA) dataset to improve the model's ability to comprehend and contextualize sports information effectively.

Existing sports QA datasets, either text-based or video-based, have limitations that hinder a comprehensive evaluation of sports understanding. Text-based datasets (bench authors, 2023; Liu et al., 2020; Xia et al., 2024a; Jardim et al., 2023) primarily assess comprehension of numerical data, rules, and context extraction, but lack detailed explanations to evaluate the underlying reasoning processes. In addition, video-based datasets, such as SportsQA (Li et al., 2024a) and SoccerNet-XFoul (Held et al., 2024b), focus on action understanding and multi-view QA, but are constrained by their narrow scope, covering only a single sport or lacking multi-level reasoning. For example, in SportsQA, questions like "What does TEAM A do before/after TEAM B's action?" mainly require recognizing sequences of actions and their outcomes, rather than connecting these actions to the underlying rules and game dynamics. This highlights the need for a comprehensive multimodal sports benchmark to evaluate the capabilities of MLLMs across a diverse range of sports, with varying levels of difficulty, to assess their ability to apply deep reasoning and rule-based understanding in real-world scenarios.

To address this gap, we introduce SPORTU, a comprehensive Sports Understanding Benchmark for sports knowledge and slow-motion multilevel video reasoning. As a multifaceted benchmark, SPORTU comprises both text and video components to facilitate a thorough assessment of models' capabilities. As illustrated in Figure 1, our dataset includes two parts: SPORTU-text and SPORTU-video. The text component, SPORTU-text, features 900 multiple-choice questions with human-annotated explanations for rule comprehension and strategy understanding. This component focuses on testing models' ability to reason about sports through question-answering, independent of visual inputs. The video component, SPORTU-video, comprises 1,701 slow-motion video clips, including 300 clips with varying camera angles and 12,048 QA pairs, categorized into three difficulty levels. The easy level is designed to be answerable without requiring sports domain knowledge, while the

hard level demands in-depth rule comprehension and accurate video perception. This tiered structure allows SPORTU-video to evaluate the sports understanding capabilities of MLLMs in a more nuanced and progressive manner. The use of slow-motion clips is crucial, as most fouls involve brief and subtle actions that may be overlooked in real-time footage. By providing models with a better opportunity to perceive and interpret these critical moments accurately, we can more effectively evaluate their performance.

To gain deeper insights into SPORTU, we evaluated 4 LLMs on SPORTU-text and 14 MLLMs on SPORTU-video, covering both open-source and proprietary models. For SPORTU-text, we selected GPT-4o, Claude-3.5-Sonnet, LLaMA-3.1, and Gemini 1.5 Pro, using few-shot learning paradigms. For SPORTU-video, we evaluated a broader range of models, including GPT-4o, Gemini 1.5 Flash, and Video-ChatGPT, using multi-frame inputs.

Key results reveal that GPT-4o achieves the highest accuracy on SPORTU-text at 71%, highlighting its relatively strong performance in text-based sports reasoning. On SPORTU-video, Qwen2-VL 72B achieves the highest overall accuracy of 70.94%, but struggles on hard-level tasks with only 44.12% accuracy. Claude-3.5-Sonnet demonstrates the best performance on hard-level tasks (52.57%). Yet, results across all models reveal significant challenges in handling complex reasoning and rule comprehension, particularly in tasks requiring deep sports knowledge. We also systematically applied different prompt strategies to evaluate reasoning performance in SPORTU-video. The results reveal that models generally achieve the highest overall performance when directly predicting the answer without providing a rationale. However, performance declines when models are required to generate a rationale first and then predict the answer. For instance, Claude-3.5-Sonnet's accuracy dropped from 52.57% with direct answer prediction to 39.32% when reasoning the rationale first. This indicates current models struggle to maintain consistency and robustness in reasoning for complex tasks. These findings highlight the need for future advancements in reasoning capabilities.

## 2 RELATED WORK

### 2.1 MULTIMODAL SPORTS ANALYSIS

Recent surveys, such as (Xia et al., 2024b), have highlighted the importance of integrating sports with NLP and CV techniques to advance research in the sports community. Traditional text-based applications have have encompassed a range of tasks, including sentiment analysis (Baca et al., 2023; Ljajić et al., 2015), game predictions (Beal et al., 2021; Xia et al., 2022; Oved et al., 2020; Tracy et al., 2023), game statistics summarization (Thomson et al., 2020; Hu et al., 2024a), and game news generation and narrative construction (Sarfati et al., 2023; Huang et al., 2020b; Wang et al., 2022a; Hu et al., 2024b). In the CV domain, studies have focused on sports action recognition (Xu et al., 2024b; Li et al., 2024c; Yuan et al., 2021; Zhu et al., 2022a), sports action quality assessment (Zahan et al., 2024; Xu et al., 2024a), and tactic analysis (He et al., 2024b). Notably, Chen et al. (2022) has demonstrated the efficacy of multimodal integration by leveraging natural language input to enhance sports videos with visualizations. While prior work has made significant strides in sports understanding, the advent of MLLMs offers a fresh perspective on this domain. In contrast to earlier multimodal approaches, MLLMs boast a distinctive blend of scalability, expressiveness, and flexibility, thereby enabling the tackling of intricate sports understanding tasks with unparalleled depth and nuance.

### 2.2 MULTIMODAL SPORTS QA

Question answering (QA) has been widely adopted to evaluate the comprehension abilities of LLMs across various domains. Several QA datasets have been introduced to evaluate models' understanding of textual information, ranging from factual recall to multi-hop reasoning (Joshi et al., 2017; Yang et al., 2018; Clark et al., 2019). In the multimodal domain, QA Benchmarks have been extended to evaluate MLLMs' image and video understanding by answering questions based on visual inputs (Fu et al., 2024; Yue et al., 2024a;b; He et al., 2024a; Zhou et al., 2024). However, sports-related QA is underrepresented in these general datasets, and even when sports topics are included, the questions often lack sufficient difficulty due to the datasets' broader focus. Existing general QA datasets tend to focus on surface-level aspects of sports, such as historical facts or well-known events in text-based tasks, and perception-level tasks, like object or action identification in video-

based tasks. As a result, they often fail to assess the deep, domain-specific knowledge and reasoning required for a nuanced understanding of sports.

Existing sports-domain-specific QA benchmarks are limited in their ability to test models' sports understanding. Text-based datasets, such as BIG-bench (bench authors, 2023), LiveQA (Liu et al., 2020), and QASports (Jardim et al., 2023), focus on factual recall, while SportQA (Xia et al., 2024a) is a notable exception, introducing rule-based questions that require scenario-based reasoning. However, even SportQA lacks explanations, which are crucial for evaluating MLLMs' reasoning processes, particularly for complex tasks. The absence of explanations limits the ability to fully assess a model's reasoning capabilities.

On the other hand, sports-domain-specific VQA datasets that combine video and text modalities to test comprehensive sports understanding are scarce. Sports-QA (Li et al., 2024a) stands out by covering eight sports, including volleyball and basketball, with videos sourced from MultiSports (Li et al., 2021) and FineGym (Shao et al., 2020). However, although it provides detailed action recognition annotations, Sports-QA does not assess models for understanding sports rules, such as foul detection. This limitation arises because MultiSports does not include foul clips, and FineGym focuses solely on granular gymnastic actions, which do not encompass rule-based scenarios.

Foul detection requires models to recognize player actions and determine rule violations, making it a critical component of sports understanding. Another benchmark, SoccerNet-XFoul (Held et al., 2024b), evaluates soccer understanding, including rule violation detection and explanation. However, its focus on a single sport limits its generalizability, as an MLLM's performance in one sport may not extend to others with different rules and dynamics. To address the limitations of previous works, we introduce SPORTU, a comprehensive benchmark that spans multiple sports and incorporates both rule-based reasoning and foul detection across text and video modalities, offering a more diverse and in-depth evaluation of sports understanding.

# 3 SPORTS UNDERSTANDING BENCHMARK

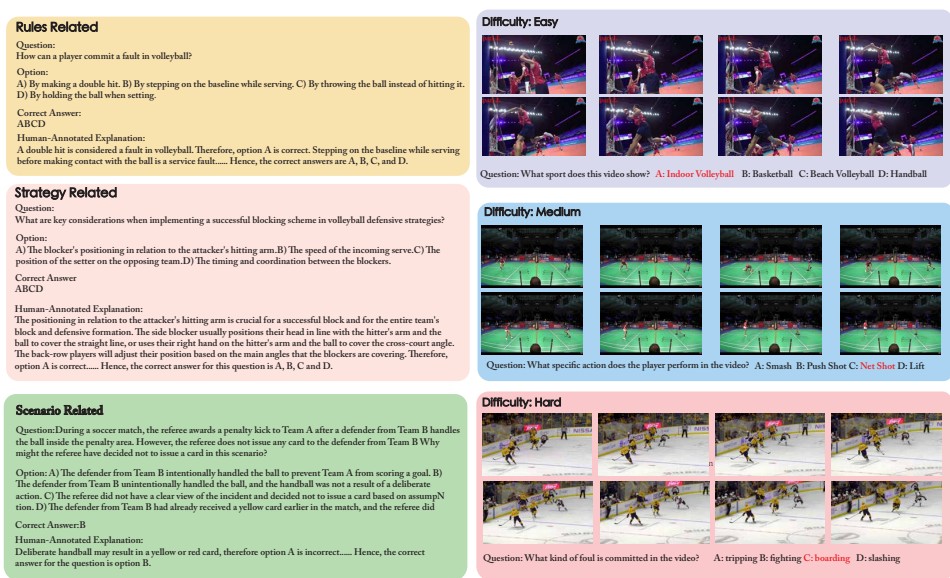

Figure 2: Examples of SPORTU-text (left) and SPORTU-video (right). Figure 1 also shows an example of an open-ended SPORTU-video question.

To address the limitations in the current sports-domain QA dataset and evaluations, we introduce SPORTU. This benchmark consists of two datasets:

**SPORTU-text** – the first pure-text-based sports QA dataset designed to evaluate models' understanding of rules, factual knowledge, scenario-based situations, and strategies with human-annotated explanations. It contains 900 QA pairs, with each question having one or more correct answers.

**SPORTU-video** – the first VQA dataset covering multiple sports, designed to evaluate MLLMs' sports understanding abilities. It features a multilevel question design using slow-motion videos, with some clips offering multiple camera angles. The tasks range from easy-level tasks, such as sports recognition, to medium-level tasks, such as Team Role Recognition, and hard-level tasks, such as rule violation explanations. The dataset contains 1,701 slow-motion video clips across 7 sports, with 12,048 QA pairs designed to test models' multi-level video-text reasoning capabilities.

Overall, SPORTU provides a comprehensive evaluation of MLLMs' ability to understand and apply sports knowledge. It fills the existing gap in current sports QA benchmarks by offering a detailed assessment across both text-based reasoning and multimodal video tasks.

## 3.1 QUALITY CONTROL

To guarantee the accuracy and consistency of annotations and question generation in SPORTU, we employed a rigorous quality verification process. Our team of annotators consisted of nine experts: two were intercollegiate student-athletes with over 12 years of experience, and seven were players who had undergone at least 5 years of training in their respective sports. It helped maintain the high standard of explanations and annotations for both SPORTU-text and SPORTU-video.

During the training phase, each team member worked with twenty examples per batch for both text-based questions and VQA tasks. They were asked to: Annotate explanations for the SPORTU-text dataset. Collect videos for SPORTU-video and annotate the key information necessary for generating questions. These key annotated variables were later used in a template-based system to generate questions. Once each annotator demonstrated full mastery of the annotation process, we officially launched the large-scale annotation phase. As part of our verification protocol, annotators were required to double-review the videos they collected to ensure accuracy and quality. We prioritized the removal of controversial or hard-to-interpret clips, especially those where even human experts might disagree or feel unsure about the decision, to minimize the risk of mislabeling.

## 3.2 SPORTU-TEXT: PURE TEXT QA

SPORTU-text is designed as the first pure-text-based sports QA dataset that provides detailed explanations for each question option, aiming to evaluate models' understanding of rules, factual knowledge, scenario-based reasoning, and strategies. The dataset consists of 900 QA pairs, with each question having one or more correct answers and accompanied by human-annotated explanations to ensure a high-quality assessment of reasoning processes. SPORTU-text can also serve as a benchmark for model explainability, allowing researchers to compare models' generated reasoning with human-provided explanations.

**Dataset Construction** Among all sports-specific QA benchmarks, only SportQA (Xia et al., 2024a) Level-3 questions assess models' deep understanding of sports knowledge by covering rule-, strategy-, and scenario-based questions. However, these questions are mixed together without clear labels for each type. To build SPORTU-text, we randomly selected 900 questions from five different sports: American football, soccer, volleyball, basketball, and tennis, and our expert annotators manually categorized the selected questions. Rule-related questions focus on explicit sports rules, strategy-related questions involve tactical or strategic decisions, and scenario-related questions provide a specific context or player interaction (e.g., "Player A performs action X, and Player B reacts"). While some questions could fit multiple categories, annotators assigned each question to the category that most accurately reflected its primary focus. Each question was carefully annotated, with detailed explanations provided for each option—whether correct or incorrect. Annotators explained why an option was correct or not, offering clear insights into the reasoning required for each answer. This additional annotation step ensures a more structured dataset and enables the evaluation of models' reasoning capabilities across distinct aspects of sports knowledge. Examples can be found in the appendix O.

## 3.3 SPORTU-VIDEO: MULTIMODAL VIDEO QA

SPORTU-video is the first multimodal video QA dataset designed to evaluate MLLMs' sports understanding across various tasks, with a specific focus on integrating visual and textual reasoning. It is unique in its use of slow-motion video clips across multiple sports and includes a multilevel

question design to test a range of model abilities, from simple recognition to complex rule-based reasoning.

SPORTU-video consists of 1,701 slow-motion video clips across 7 different sports, including soccer, basketball, volleyball, ice hockey, tennis, baseball, and badminton. We also ensured that for some sports, the videos featured multiple camera angles to challenge the models' ability to capture consistent judgments across different perspectives. Our expert annotators manually cropped video clips from replay footage to include multiple perspectives of the same foul, as such replays are standard in sports broadcasts. This process ensures accuracy while minimizing additional manual work. Specifically, 300 video scenes were selected to include multi-angle views. Each clip is accompanied by one or more QA pairs, for a total of 12,048 QA pairs (with 10,973 Multiple Choice Questions and 1,075 open-ended questions based on the explanations that the annotators labeled), with three levels of complexity: Easy: 25.36%, Medium: 50.22%, Hard: 24.42%.

**Dataset Construction** The construction of SPORTU-video began by identifying the types of tasks that could be asked based on the sports domain. Some questions, such as sports recognition and identifying the number of players, are common across all sports. Other questions are sports-specific, requiring knowledge of rules or strategies for each sport. Full question templates can be found in the appendix N. To classify the questions, we considered three different difficulty levels based on the required sports knowledge and reasoning. **Easy-level questions:** These tasks rely on commonsense, such as basic sports recognition. For example, questions like "What sport does the video show?" or "How many players are shown?" do not require sports knowledge. **Medium-level questions:** These questions require sports knowledge beyond commonsense. For example, models are asked to identify which team is on offense based on the video or to recognize specific roles, such as a libero in volleyball, by identifying the libero's jersey color. **Hard-level questions:** These tasks involve deep rule-based reasoning. For example, identifying rule violations or technical errors requires models to understand the sport's rules in detail and apply them to the specific context of the video.

Once the question types were defined, the annotators collected the appropriate video clips and labeled the corresponding ground truths that the videos showed, as multiple questions could often be answered from a single video. There is no restriction on the data source for these slow-motion clips as long as the annotators follow the original copyright and license requirements, and each video was manually cropped to ensure high-quality footage suitable for detailed action analysis and rule comprehension. For multiple-choice questions, annotators labeled the ground truth category, such as the specific foul ("handball" or "offsid"), which was then used to generate multiple-choice questions with distractors derived from other categories. For open-ended questions, annotators provided detailed explanations for the rule violation or foul observed in the video. These explanations were used as the ground truth for generating open-ended questions, allowing models to be tested on reasoning and explanation generation.

**Explanations** One of the unique aspects of SPORTU-video is its emphasis on tasks involving foul detection and technical errors, where open-ended questions are accompanied by detailed human-annotated explanations. These explanations provide insights into why certain fouls or errors occurred, helping to evaluate models not only on their accuracy but also on their ability to explain the reasoning behind their answers. Due to limited annotation resources, these explanations are only provided for the most challenging tasks involving rule violations and technical errors, as these require models to combine action recognition with textual rule understanding to determine the violation. We believe that these tasks offer the most rigorous test of a model's ability to connect sports knowledge with video comprehension.

## 4 EXPERIMENT

We compare MLLM's performance on the SPORTU benchmark, We also evaluate the ability of models to produce explanations. We start by describing the MLLMs in our experiments and their experimental settings(§4.1), followed by prompting strategies (§4.2), and evaluation metrics (§4.3).

### 4.1 MODELS

**SPORTU-text Evaluation:** We evaluated several leading language models on the SPORTU-text, including open-source models like Llama-3.1-405B (Dubey et al., 2024), and closed-source models

such as GPT-4o (2024-08-06 version) (OpenAI, 2024a), Gemini-1.5 Pro(Gemini Team, 2024a), and Claude-3.5-Sonnet (20240620 version) (Anthropic, 2024b). Access to these models was facilitated through their respective APIs.

**SPORTU-video Evaluation:** We investigate a range of MLLMs, including 6 close-source models and 6 open-source models. For close-source models, we evaluated GPT-4o (2024-08-06 version) and -4omini (OpenAI, 2024a), Gemini 1.5 Pro (Gemini Team, 2024a), Gemini 1.5 Flash (Gemini Team, 2024b), Claude-3.5-Sonnet (20240620 version) (Anthropic, 2024b) and-3.0-Haiku Anthropic (2024).For open-source models, we evaluated ChatUniVi (Jin et al., 2023), LLaVA-NeXT (Liu et al., 2024), mPLUG-Owl3 (Ye et al., 2024), Tarsier (Wang et al., 2024), Video-ChatGPT (Maaz et al., 2024), VideoChat2(Li et al., 2024b), ST-LLM (Liu et al., 2025), and Qwen2-VL-72B (Bai et al., 2023). For the closed-source models, we adhered to the default settings provided by their official APIs. GPT and Claude family models processed ten image frames extracted from the video content as input. The Gemini family models processed the entire video, as their API supports video input. Due to computing resource limitations, we used the 7B versions of all open-source models in this evaluation except Qwen2-VL-72B, which can be accessed by API. For VideoChat, we set 'max_frames' to 100, while for the other open-source models, we used 16 frames as input. Across all closed and open-source models, we set the temperature parameter to 0 to ensure consistent response generation. All inferences are run on an NVIDIA RTX A6000.

## 4.2 PROMPTING STRATEGIES

We apply three different prompting strategies to generate answers and/or explanations. We represent the input as X for the question and answer options, Y for the answer, and R for the explanation (rationale). Our three strategies are:

- $X \to Y$: No-CoT, which directly predicts the answer.
- $X \to RY$: Reasoning where answer inference is conditioned to the rationale. This strategy asks the model to engage in step-by-step reasoning first, and then answer the question. This approach is based on chain of thought (CoT) prompting, which has been shown to improve LLMs' prediction accuracy across various reasoning tasks Wei et al. (2022). The zero-shot CoT method is adapted from Kojima et al. (2022). We prompt the model to 'think step by step, then provide the correct answer.'
- $X \to YR$: This strategy asks the model to answer the question first, followed by the rationale for why the model chose that option. This prompting method has proven effective on REV Chen et al. (2023).

For the SPORTU-text evaluation, LLMs have demonstrated their capability for in-context learning by utilizing exemplars through few-shot prompting (Brown, 2020). Therefore, we use four prompting baselines for evaluation: zero-shot $X \to Y$ (0S), zero-shot $X \to RY$ (0CoT), five-shot $X \to Y$ (5S), and five-shot $X \to RY$ (5CoT). For the five-shot method, we provide five exemplars with only the answers. The five exemplars for the five-shot CoT method include both the answers and human-annotated rationales.

For the SPORTU-video evaluation, we use all three prompting methods. The zero-shot $X \to YR$ prompting method is applied to conduct human error analysis. More details of models' specific prompts are shown in the appendix L.

## 4.3 EVALUATION METRICS

We use accuracy (prediction compared to ground truth) to evaluate the predictions of each model in multiple-choice tasks. We explore several methods to evaluate different aspects of model-generated explanations and open-ended questions: ROUGE-L (Lin, 2004), BERTScore (Zhang et al., 2019), BLEURT (Sellam et al., 2020), CTC-Preservation (Deng et al., 2021), and G-Eval Score (Liu et al., 2023). ROUGE-L computes the surface-form similarity between model-generated explanations and reference (human-annotated) explanations. BERTScore and BLEURT measure semantic similarity using pre-trained BERT (Devlin, 2018) and fine-tuned BERT models respectively. CTC metrics evaluate the information alignment of model-generated explanations. G-Eval is a framework that uses large language models with chain-of-thought reasoning and a form-filling paradigm to assess

the quality of NLG outputs. We apply this framework to evaluate the accuracy, conciseness, and relevance of the model-generated explanations compared to the ground truth, reporting these as an overall score. Since LLMs might favor their own answers, we utilize all four models evaluated in the SPORTU-text part to implement the G-Eval process for SPORTU-text results and three models (excluding LLama3.1-405B) for the SPORTU-video part. We also calculate the average score of the models to obtain an overall score. The detailed prompt can be found in the Appendix J. To ensure the reliability of the G-Eval scores, we conducted a human evaluation on a randomly selected set of 20 questions per sport across 7 sports, totaling 140 questions. This subset was used to compare the G-Eval scores with human-annotated scores across 14 models. By using the same set of questions for all models, we ensured consistency in comparison, allowing us to assess whether the G-Eval scores were in line with human judgment. The human annotators rated the model-generated content using the same criteria as G-Eval, providing a reference point for how well the automated evaluation aligns with human evaluation.

## 5  RESULTS

We compared the performance of different models across both SPORTU-text and SPORTU-video.

Overall results for SPORTU-text are shown in Table 1 and Table 2, while the multiple-choice QA results for SPORTU-video are presented in Table 3, and the open-ended QA results are displayed in Table 4. Among all models, GPT-4o performs the best in SPORTU-text, achieving the highest accuracy of 71% in the five-shot CoT prompt setting, along with a G-Eval score of 4.16 in the zero-shot CoT

Table 1: Performance of LLMs with Standard Prompt Settings

| Setting | Model | Acc.(%) |
|---|---|---|
| $X{\rightarrow}Y(0S)$ | Claude-3.5-Sonnet | 64.33 |
| | gemini-1.5 Pro | 63.33 |
| | GPT-4o | 70.22 |
| | Llama3.1-405B | 66.67 |
| $X{\rightarrow}Y(5S)$ | Claude-3.5-Sonnet | 68.44 |
| | gemini-1.5 Pro | 64.11 |
| | GPT-4o | 70.78 |
| | Llama3.1-405B | 66.33 |

prompt setting. The G-Eval score confirms that GPT-4o can somewhat grasp sports-related rules, strategies, and scenario-based questions. However, there remains a notable gap compared to expert performance, which exceeds 90%, as mentioned in Xia et al. (2024a).

Table 2: Performance of LLMs on SPORTU-text evaluation across CoT settings. Metrics include: Accuracy (Acc), ROUGE-L (R-L), BERTScore (B-S), BLEURT (BL), CTC Preservation (CTC), GPT-based G-Eval (G-E), Gemini-based Eval (GEM), Claude-based Eval (CL), Llama-based Eval (LL), and Average G-Eval score (AVG). GEM uses Gemini 1.5 pro, CL uses Claude-3.5-Sonnet, and LL uses Llama3.1-405B for evaluation.

| Setting | Model | Acc(%) | R-L | B-S | BL | CTC | G-E | GEM | CL | LL | AVG |
|---|---|---|---|---|---|---|---|---|---|---|---|
| $X{\rightarrow}RY(0CoT)$ | Claude-3.5-Sonnet | 64.67 | 0.26 | 0.65 | 0.57 | 0.43 | 3.78 | 3.25 | 3.28 | 4.07 | 3.60 |
| | gemini-1.5 Pro | 62.67 | 0.28 | 0.62 | 0.53 | 0.43 | 3.79 | 3.57 | 3.39 | 3.98 | 3.68 |
| | GPT-4o | 68.78 | 0.27 | 0.66 | 0.57 | 0.43 | 4.16 | 3.42 | 3.43 | 4.37 | 3.85 |
| | Llama3.1-405B | 64.44 | 0.25 | 0.64 | 0.55 | 0.43 | 3.89 | 3.19 | 2.74 | 3.90 | 3.39 |
| $X{\rightarrow}RY(5CoT)$ | Claude-3.5-Sonnet | 65.22 | 0.27 | 0.65 | 0.56 | 0.43 | 3.98 | 3.43 | 3.39 | 4.15 | 3.74 |
| | gemini-1.5 Pro | 61.22 | 0.30 | 0.62 | 0.53 | 0.43 | 3.73 | 3.51 | 3.49 | 3.38 | 3.53 |
| | GPT-4o | 71.00 | 0.33 | 0.68 | 0.58 | 0.44 | 4.13 | 3.52 | 3.59 | 4.15 | 3.85 |
| | Llama3.1-405B | 65.22 | 0.32 | 0.67 | 0.57 | 0.44 | 3.81 | 3.28 | 3.33 | 4.02 | 3.61 |

For the SPORTU-video multiple-choice task, Qwen2-VL-72B achieved the highest overall accuracy at 70.94% on the $X \rightarrow Y$ setting, followed by Claude-3.5-Sonnet (70.18%). Models tend to perform well on easy-level questions but show a significant gap in hard-level questions, indicating a lack of domain knowledge, particularly in rule comprehension. For example, GPT-4o leads on the hard-level tasks with only 57.84%. Among the open-source models, LLAVA-NeXT outperformed others.

By comparing three different prompting strategies, we observed that $X \rightarrow Y$ achieved the highest overall performance across most models, outperforming both $X \rightarrow YR$ and $X \rightarrow RY$. For most

Table 3: Overall performance of MLLMs on SPORTU-video for multiple-choice questions. The best results are **bolded**. The results highlight that models perform best with the $X \rightarrow Y$, followed by $X \rightarrow YR$, and $X \rightarrow RY$.

| Model | Accuracy(%) | | |
|---|---|---|---|
| | X-YR | X-RY | X-Y |
| Close-source Model | | | |
| Claude-3.0-Haiku | **48.07** | 47.19 | 47.95 |
| Claude-3.5-Sonnet | 69.52 | 55.08 | **70.18** |
| Gemini 1.5 Pro | **65.13** | 63.04 | 64.93 |
| Gemini 1.5 Flash | 59.97 | 46.68 | **62.52** |
| GPT-4omini | 57.24 | 42.06 | **58.19** |
| GPT-4o | 68.00 | 65.56 | **68.79** |
| Open-source Model | | | |
| ChatUniVi | **42.35** | 32.58 | 41.89 |
| LLaVA-NeXT | **68.89** | 62.16 | 63.72 |
| mPLUG-Owl3 | 59.26 | **61.27** | 60.80 |
| ST-LLM | 41.59 | 40.09 | **46.39** |
| Tarsier | **61.32** | 55.70 | 60.99 |
| Video-ChatGPT | **44.63** | 42.36 | 34.05 |
| VideoChat2 | 61.55 | **62.79** | 61.53 |
| Qwen2-VL-72B | 69.18 | 62.65 | **70.94** |

models, the accuracy follows the order: $X \rightarrow Y > X \rightarrow YR > X \rightarrow RY$. A detailed comparison of the prompting strategies across different difficulty levels is provided in Appendix G.

We also found that when models generated the rationale first (in $X \rightarrow RY$), the final answer prediction was often influenced by incorrect or hallucinated reasoning processes. Examples illustrating these errors can be found in Appendix H. This observation aligns with the findings of Zhang et al. (2023), further emphasizing the challenges of reasoning-based approaches in complex tasks.

For open-ended questions, Close source models, along with the Qwen2-VL-72B model, achieved higher G-Eval and human rating scores compared to the 7B open-source models. This indicates that close-source models exhibit stronger reasoning abilities than the 7B open-source models. GPT-4o again led with a G-Eval score of 1.84, a result further validated by human annotators. Even with human evaluations, the score closely aligned with G-Eval's assessment, with the model not exceeding a score of 3 when evaluated using all three LLMs. A score of 1 indicates very poor performance, and 2 suggests poor performance based on the evaluation criteria. This highlights the model's struggle to connect observed actions with relevant domain knowledge, such as identifying technical errors or specific rules. Overall, none of the MLLMs achieved an average score above 3, demonstrating a gap in deep domain knowledge required for video sports understanding. We also noticed that among the evaluated metrics, G-Eval demonstrates the closest alignment with human ratings, with a Pearson correlation coefficient of 0.41. However, as this and other metrics exhibit low correlations with human ratings, it highlights the need for developing a domain-specific metric for evaluating sports content in the future. More can be found in Appendix I.

Additionally, we notice that the performance of models on multi-angle videos shows variability depending on the camera perspectives for the same scene, indicating that models struggle with consistent understanding across different camera angles. More details can be found in the appendix D.

## 5.1 ERROR ANALYSIS

To gain deeper insights into the limitations of MLLMs, we applied the X→YR prompting method, where models first generated an answer and then explained their reasoning. This provided valuable information about how models approached reasoning, especially in complex tasks like foul detection. We analyzed errors across both open-ended and multiple-choice tasks, selecting 20 incorrect examples per sport for each model, resulting in a total of 3920 errors.

Figure 3 shows the radar chart representing the distribution of different error types. The most frequent error was Question Understanding Error, particularly in questions like "Why is it a foul in the video?" where models incorrectly responded that there was no foul despite the question's clear presupposition. Another common issue was Hallucination Error, such as when a model mentioned a referee in its explanation, even though no referee was visible in the video. Detailed examples of these errors and further case studies are provided in Appendix M.

Table 4: Model performance on SPORTU-video open-ended tasks. Metrics include ROUGE-L (R-L), BERTScore (B-S), BLEURT (BL), CTC Preservation (CTC), GGPT-based G-Eval (G-E), Gemini-based Eval (GEM), Claude-based Eval (CL), Llama-based Eval (LL), and Average G-Eval score (AVG). GEM uses Gemini 1.5 pro, CL uses Claude-3.5-Sonnet, and LL uses Llama3.1-405B for evaluation and Human Rating (H-R*). * denotes human ratings conducted to verify the reliability of the G-Eval scores as mentioned in 4.3

| Model | R-L | B-S | BL | CTC | G-E | GEM | CL | AVG | H-R* |
|---|---|---|---|---|---|---|---|---|---|
| Close-source Models | | | | | | | | | |
| Claude-3.0-Haiku | 0.08 | 0.41 | 0.43 | 0.39 | 1.55 | 1.80 | 1.63 | 1.66 | 1.93 |
| Claude-3.5-Sonnet | 0.05 | 0.40 | 0.43 | 0.39 | 1.62 | 1.89 | 1.59 | 1.70 | 2.13 |
| Gemini 1.5 Pro | 0.08 | 0.38 | 0.36 | 0.38 | 1.11 | 1.16 | 1.20 | 1.16 | 1.19 |
| Gemini 1.5 Flash | 0.13 | 0.45 | 0.42 | 0.39 | 1.34 | 1.70 | 1.62 | 1.55 | 1.84 |
| GPT-4omini | 0.05 | 0.39 | 0.36 | 0.38 | 1.60 | 1.94 | 1.65 | 1.73 | 2.17 |
| GPT-4o | 0.07 | 0.41 | 0.43 | 0.39 | 1.84 | 1.17 | 1.75 | 1.59 | 2.51 |
| Open-source Models | | | | | | | | | |
| ChatUniVi | 0.07 | 0.39 | 0.37 | 0.38 | 1.27 | 1.39 | 1.45 | 1.37 | 1.48 |
| LLaVA-NeXT | 0.17 | 0.47 | 0.38 | 0.40 | 1.47 | 1.63 | 1.75 | 1.61 | 1.62 |
| mPLUG-Owl3 | 0.15 | 0.44 | 0.37 | 0.39 | 1.38 | 1.60 | 1.75 | 1.58 | 1.46 |
| Tarsier | 0.12 | 0.45 | 0.36 | 0.40 | 1.36 | 0.70 | 1.78 | 1.28 | 1.63 |
| Video-ChatGPT | 0.08 | 0.39 | 0.35 | 0.38 | 1.08 | 1.11 | 1.36 | 1.19 | 1.22 |
| VideoChat2 | 0.23 | 0.49 | 0.35 | 0.40 | 1.43 | 1.73 | 1.79 | 1.65 | 1.48 |
| ST-LLM | 0.13 | 0.38 | 0.20 | 0.41 | 1.30 | 1.50 | 1.52 | 1.44 | 1.22 |
| Qwen2-VL-72B | 0.10 | 0.42 | 0.39 | 0.41 | 1.62 | 1.94 | 1.72 | 1.76 | 2.08 |

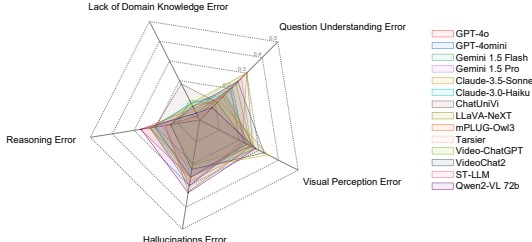

Figure 3: Error type distribution across different MLLMs on SPORTU-video tasks. The analysis reveals that Question Understanding Error is the most common issue, followed by Hallucination Error. Each error type highlights specific model limitations in comprehending the task.

## 6  CONCLUSION

In this paper, we introduced SPORTU, a benchmark designed to evaluate the sports understanding capabilities of Multimodal Large Language Models (MLLMs). SPORTU comprises two components: SPORTU-text, which assesses models' comprehension of rules, strategies, and scenarios through multiple-choice questions, and SPORTU-video, which evaluates their ability to apply this knowledge to real-world sports footage, including tasks like recognition, foul detection, and rule application. By integrating both text and video tasks, SPORTU provides a holistic assessment of reasoning abilities across different levels of complexity. Our results reveal that while models like GPT-4o show progress in text-based reasoning, they struggle with scenario-based reasoning and connecting visual actions with domain-specific rules. Error analysis highlights issues such as question misunderstanding and hallucination, emphasizing the need for improved reasoning capabilities in future models. We also discuss the broader impacts of our findings in Appendix B. We hope SPORTU will inspire advancements in MLLMs and contribute to robust real-world sports understanding.

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

CONTENTS

## A  MORE RELATED WORKS

In this section, we provide a detailed comparison table 5 of various QA datasets across multiple dimensions. This includes QA Type, whether explanations are provided, the presence of multilevel difficulty, and the coverage of domain knowledge in terms of both action and rule. Additional features compared include whether datasets incorporate slow motion, multi-camera angles, and the number of sports covered. Average explanation or answer length refers to the average word count for open-ended explanations or answers.

Table 5: Comparison of various QA datasets across multiple dimensions. MC means Multiple Choice. OE means open-ended. Avg Exp. and Ans. Length refers to the average word count for open-ended explanations and answers.

| Benchmark | QA Type | Question Type | Explanation | Multilevel Diff. | Domain Knowledge | | Slow Motion | Multi Camera Angle | No. of Sports | Avg Exp. or Ans. Length |
| --- | --- | --- | --- | --- | --- | --- | --- | --- | --- | --- |
| | | | | | Action | Rule | | | | |
| Text | | | | | | | | | | |
| BIG-bench (bench authors, 2023) | Context-free | MC | ✗ | ✗ | ✗ | ✗ | - | - | 5 | - |
| LiveQA (Liu et al., 2020) | Context extractive | MC | ✗ | ✗ | ✗ | ✗ | - | - | 1 | - |
| QASports (Jardim et al., 2023) | Context extractive | OE | ✗ | ✗ | ✗ | ✗ | - | - | 3 | - |
| SportQA (Xia et al., 2024a) | Context-free | MC | ✗ | ✓ | ✗ | ✓ | - | - | 35 | - |
| **SPORTU-text (Ours)** | Context-free | MC | ✓ | ✓ | ✗ | ✓ | - | - | 5 | 100.19 |
| Video | | | | | | | | | | |
| Sports-QALi et al. (2024a) | Video QA | OE | ✗ | ✗ | ✓ | ✗ | ✗ | ✗ | 8 | - |
| SoccerNet-XFoul (Held et al., 2024b) | Video QA | OE | ✓ | ✗ | ✓ | ✓ | ✗ | ✓ | 1 | 25 |
| **SPORTU-video (Ours)** | Video QA | OE + MC | ✓ | ✓ | ✓ | ✓ | ✓ | ✓ | 7 | 13.29 |

## B  BROADER IMPACTS

Sports understanding is a critical domain for MLLMs to develop as they are increasingly applied to real-world tasks, such as supporting sports education. MLLMs with advanced reasoning capabilities can empower non-experts to quickly grasp the rules and dynamics of sports, enhancing their ability to enjoy and understand games. They also hold promise for supporting advanced applications like sports strategy analysis and real-time decision-making, which brings new chapters for the sports community.

The SPORTU benchmark addresses key gaps in existing datasets by systematically evaluating MLLMs' sports understanding across three difficulty levels: easy, medium, and hard. This tiered structure reveals how models perform well on commonsense-based questions but face challenges as tasks demand deeper sports knowledge and reasoning. These findings highlight the current limitations of MLLMs in sports reasoning and underscore the need for further advancements to address these gaps. As the timing when we write the paper, the MLLMs are still struggling with combining actions with corresponding rules and explaining the rationale, so we believe that SPORTU can meaningfully guide the development and benchmarking of future MLLMs in the sports domain.

## C  DISCUSSION

At the time of writing this paper, current SOTA MLLMs still perform poorly on challenging tasks that require combining sports knowledge with corresponding actions, particularly in connecting these actions to various rules. Through different prompting strategies, we found that the models performed even worse when required to provide a reasoning process before predicting the final answer. However, the sports domain necessitates that MLLMs have a robust and reliable reasoning process, as explaining the concept behind the questions, where people can be inspired and learn from the context, is often more important than simply providing the final result.

Additionally, we observed that better frame extraction strategies need to be developed specifically for sports tasks to ensure that the actions critical for reasoning about the question are clearly provided to the model. A more effective grounding method could also enhance the model's ability to distinguish actions more accurately. During the experiments, we noticed that while models sometimes captured and described most of the correct movements, they often failed to infer the corresponding rule violations. This highlights a significant gap in connecting recognized actions to specific rules, underscoring the need for models to not only identify movements but also to understand what a rule violation should look like.

We hope that our benchmark can positively contribute to the community, serving as a foundational step to bring more attention to the sports domain and to inspire the development of advanced models that can help people engage more deeply with sports.

# D MULTI-ANGLE VIDEOS RESULT

## D.1 MODEL PERFORMANCE

Table 6: Accuracy comparison across multiple camera angles of the same question

| Model (%) | P(All Correct) | P(All False) | P(At least One Correct & At least One False) |
|---|---|---|---|
| Claude-3.0-Haiku | 32.17 | 32.11 | 35.72 |
| Claude-3.5-Sonnet | 52.86 | 18.59 | 28.55 |
| Gemini 1.5 Pro | 43.24 | 25.41 | 31.35 |
| Gemini 1.5 Flash | 48.48 | 22.96 | 28.55 |
| GPT-4omini | 36.89 | 28.40 | 34.72 |
| GPT-4o | 47.42 | 20.84 | 31.73 |
| ChatUniVi | 29.31 | 38.52 | 32.17 |
| LLaVA-NeXT | 53.67 | 22.26 | 24.07 |
| mPLUG-Owl3 | 42.77 | 27.91 | 29.31 |
| ST-LLM | 22.72 | 58.39 | 24.88 |
| Tarsier | 48.25 | 26.05 | 25.70 |
| Video-ChatGPT | 25.06 | 35.84 | 39.10 |
| VideoChat2 | 47.84 | 21.27 | 30.89 |
| Qwen2-VL-72B | 51.78 | 20.11 | 28.11 |

In this section, we analyze the performance of models when answering the same question from different camera angles within the same scene. The objective is to evaluate how consistently models understand the same scenario when presented from multiple perspectives. For example, if the question is "What color is the person's jersey?", the same question is asked across different camera angles for the same scene. The results are categorized into three distinct cases:

- P(All Correct): This indicates the percentage of instances where the model answered correctly for all camera angles of the same scene. A higher percentage reflects the model's ability to consistently interpret and understand the scene across multiple perspectives.

- P(All False): This shows the percentage of instances where the model answered incorrectly for all angles of the same scene, highlighting consistent misunderstanding across different perspectives.

- P(At least One Correct& At least One False): This category reflects situations where the model answered correctly for at least one camera angle but incorrectly for at least one other, indicating that the model's understanding varies depending on the perspective.

The analysis reveals that camera angle variation can significantly affect the model's performance. While some models are more consistent across angles, others show a noticeable drop in accuracy when the angle changes, suggesting challenges in maintaining robust understanding across multiple viewpoints. This highlights a critical area for further research and improvement in multimodal models when handling multi-angle video inputs. We will present an example in the next section where the model answers the question incorrectly when viewed from one camera angle but answers it correctly from another.

## D.2 Example of Multi-Angle Videos

In this section, we provide an example that shows that under the same question, the model got different answers from two different camera angles.

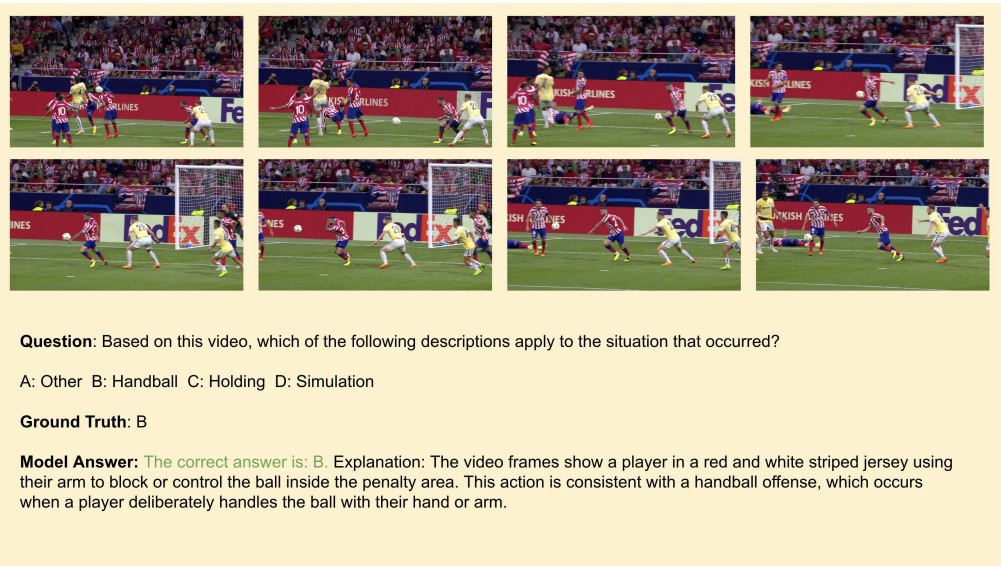

**Question**: Based on this video, which of the following descriptions apply to the situation that occurred?

A: Other  B: Handball  C: Holding  D: Simulation

**Ground Truth**: B

**Model Answer:** The correct answer is: B. Explanation: The video frames show a player in a red and white striped jersey using their arm to block or control the ball inside the penalty area. This action is consistent with a handball offense, which occurs when a player deliberately handles the ball with their hand or arm.

Figure 4: Under camera angle one, the model answers the question correctly.

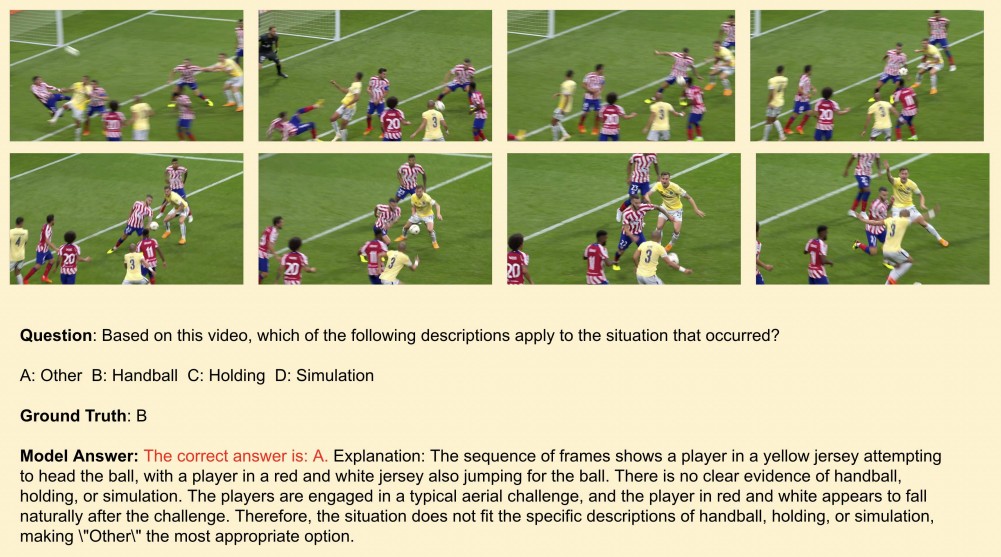

**Question**: Based on this video, which of the following descriptions apply to the situation that occurred?

A: Other  B: Handball  C: Holding  D: Simulation

**Ground Truth**: B

**Model Answer:** The correct answer is: A. Explanation: The sequence of frames shows a player in a yellow jersey attempting to head the ball, with a player in a red and white jersey also jumping for the ball. There is no clear evidence of handball, holding, or simulation. The players are engaged in a typical aerial challenge, and the player in red and white appears to fall naturally after the challenge. Therefore, the situation does not fit the specific descriptions of handball, holding, or simulation, making \"Other\" the most appropriate option.

Figure 5: Under camera angle two, the model answers the question wrong.

# E ZERO SHOT BASED PROMPT SETTING RESULTS ACROSS RULES-, STRATEGY-, AND SCENARIO-RELATED QUESTION ON SPORT-TEXT

## E.1 ZERO SHOT RESULT

Table 7: Zero Shot Performance of LLMs across different question types.

| Model(%) | Acc. (overall) | Acc. (rule) | Acc. (scenario) | Acc. (strategy) |
|---|---|---|---|---|
| Claude-3.5-Sonnet | 64.33 | 58.27 | 67.42 | 70.62 |
| GPT-4o | 70.22 | 68.56 | 69.69 | 74.58 |
| Llama3.1-405B | 66.67 | 62.33 | 69.97 | 68.93 |
| Gemini-1.5 Pro | 63.33 | 60.43 | 64.41 | 67.05 |

## E.2 ZERO SHOT WITH CoT RESULT

Table 8: Zero Shot CoT Performance of LLMs across different Question Types.

| Model | Acc.(%) | ROUGE-L | BERTScore | BLEURT | CTC Presv. | G-Eval |
|---|---|---|---|---|---|---|
| **Overall** | | | | | | |
| Claude-3.5-Sonnet | 64.67 | 0.26 | 0.65 | 0.57 | 0.43 | 3.78 |
| Gemini-1.5 | 62.67 | 0.28 | 0.62 | 0.53 | 0.43 | 3.79 |
| GPT-4o | 68.78 | 0.27 | 0.66 | 0.57 | 0.43 | 4.16 |
| Llama3.1-405B | 64.44 | 0.25 | 0.64 | 0.55 | 0.43 | 3.89 |
| **Rule** | | | | | | |
| Claude-3.5-Sonnet | 64.67 | 0.26 | 0.65 | 0.57 | 0.43 | 3.79 |
| Gemini-1.5 | 62.67 | 0.28 | 0.62 | 0.53 | 0.42 | 3.80 |
| GPT-4o | 68.78 | 0.27 | 0.66 | 0.57 | 0.43 | 4.16 |
| Llama3.1-405B | 64.44 | 0.25 | 0.64 | 0.55 | 0.43 | 3.89 |
| **Strategy** | | | | | | |
| Claude-3.5-Sonnet | 67.43 | 0.25 | 0.65 | 0.57 | 0.43 | 3.89 |
| Gemini-1.5 | 62.29 | 0.26 | 0.61 | 0.52 | 0.42 | 3.98 |
| GPT-4o | 73.14 | 0.27 | 0.66 | 0.56 | 0.43 | 4.23 |
| Llama3.1-405B | 65.71 | 0.24 | 0.63 | 0.54 | 0.43 | 3.98 |
| **Scenario** | | | | | | |
| Claude-3.5-Sonnet | 66.20 | 0.26 | 0.65 | 0.57 | 0.43 | 3.86 |
| Gemini-1.5 | 61.13 | 0.27 | 0.62 | 0.53 | 0.43 | 3.71 |
| GPT-4o | 68.73 | 0.27 | 0.66 | 0.56 | 0.43 | 4.10 |
| Llama3.1-405B | 66.57 | 0.25 | 0.64 | 0.55 | 0.43 | 3.91 |

## F  FIVE SHOT BASED PROMPT SETTING RESULTS ACROSS RULES-, STRATEGY-, AND SCENARIO-RELATED QUESTION ON SPORT-TEXT

### F.1  FIVE SHOT RESULT

Table 9: Five Shot Performance of LLMs across different question types.

| Model (%) | Acc. (overall) | Acc. (rule) | Acc. (scenario) | Acc. (strategy) |
|---|---|---|---|---|
| Claude-3.5-Sonnet | 68.44 | 65.04 | 70.82 | 70.62 |
| GPT-4o | 70.78 | 70.46 | 70.54 | 71.75 |
| Llama3.1-405B | 66.33 | 62.6 | 68.84 | 68.93 |
| Gemini-1.5 Pro | 64.11 | 63.14 | 61.76 | 70.62 |

### F.2  FIVE SHOT WITH COT RESULT

Table 10: Five Shot CoT Performance of LLMs across different question types.

| Model | Acc.(%) | ROUGE-L | BERTScore | BLEURT | CTC Presv. | G-Eval |
|---|---|---|---|---|---|---|
| **Overall** | | | | | | |
| Claude-3.5-Sonnet | 65.22 | 0.27 | 0.65 | 0.56 | 0.43 | 3.98 |
| Gemini-1.5 | 61.22 | 0.30 | 0.62 | 0.53 | 0.43 | 3.73 |
| GPT-4o | 71.00 | 0.33 | 0.68 | 0.58 | 0.44 | 4.13 |
| Llama3.1-405B | 65.22 | 0.32 | 0.67 | 0.57 | 0.44 | 3.81 |
| **Rule** | | | | | | |
| Claude-3.5-Sonnet | 64.77 | 0.27 | 0.65 | 0.56 | 0.43 | 3.95 |
| Gemini-1.5 | 59.89 | 0.31 | 0.63 | 0.54 | 0.43 | 3.63 |
| GPT-4o | 70.46 | 0.34 | 0.69 | 0.58 | 0.44 | 4.09 |
| Llama3.1-405B | 62.60 | 0.33 | 0.67 | 0.58 | 0.44 | 3.73 |
| **Strategy** | | | | | | |
| Claude-3.5-Sonnet | 66.86 | 0.26 | 0.65 | 0.56 | 0.43 | 4.15 |
| Gemini-1.5 | 63.84 | 0.29 | 0.62 | 0.53 | 0.42 | 3.99 |
| GPT-4o | 71.43 | 0.33 | 0.68 | 0.58 | 0.44 | 4.23 |
| Llama3.1-405B | 66.29 | 0.32 | 0.67 | 0.57 | 0.44 | 3.89 |
| **Scenario** | | | | | | |
| Claude-3.5-Sonnet | 64.79 | 0.27 | 0.65 | 0.57 | 0.43 | 3.95 |
| Gemini-1.5 | 61.19 | 0.30 | 0.62 | 0.53 | 0.43 | 3.75 |
| GPT-4o | 71.27 | 0.32 | 0.68 | 0.58 | 0.44 | 4.12 |
| Llama3.1-405B | 67.61 | 0.32 | 0.67 | 0.57 | 0.44 | 3.87 |

# G   RESULT OF THREE PROMPT SETTINGS ON SPORTU-VIDEO ACROSS DIFFERENT LEVELS OF DIFFICULTY

Table 11: Overall performance of MLLMs on SPORTU-video for multiple-choice questions across three difficulty levels. The best results are **bolded**. The results highlight that models perform best with the $X \to Y$ prompt (25/56 leading performances), followed by $X \to$ (21/56), and $X \to YR$ (10/56).

| Model | Difficulty | Performance | | |
|---|---|---|---|---|
| | | **X-YR** | **X-RY** | **X-Y** |
| Claude-3.0-Haiku | Easy | **68.41** | 66.58 | 66.62 |
| | Medium | 46.43 | 46.11 | **46.53** |
| | Hard | 20.12 | 18.94 | **22.42** |
| | Overall | **48.07** | 47.19 | 47.95 |
| Claude-3.5-Sonnet | Easy | 88.65 | 63.83 | **89.15** |
| | Medium | 65.10 | 55.52 | **65.88** |
| | Hard | 52.57 | 39.32 | **53.06** |
| | Overall | 69.52 | 55.08 | **70.18** |
| Gemini 1.5 Pro | Easy | 85.85 | 85.20 | **87.52** |
| | Medium | 58.25 | 58.11 | **61.22** |
| | Hard | **43.53** | 42.75 | 39.98 |
| | Overall | **65.13** | 63.04 | 64.93 |
| Gemini 1.5 Flash | Easy | **85.99** | 59.07 | 85.19 |
| | Medium | 53.38 | 50.85 | **58.56** |
| | Hard | **38.73** | 12.89 | 38.26 |
| | Overall | 59.97 | 46.68 | **62.52** |
| GPT-4omini | Easy | 66.09 | 59.02 | **66.68** |
| | Medium | 55.69 | 42.25 | **58.12** |
| | Hard | **47.54** | 13.67 | 44.49 |
| | Overall | 57.24 | 42.06 | **58.19** |
| GPT-4o | Easy | **84.89** | 79.92 | 84.30 |
| | Medium | 62.31 | **65.98** | 64.83 |
| | Hard | **57.84** | 40.51 | 56.20 |
| | Overall | 68.00 | 65.56 | **68.79** |
| Qwen2-VL-72B | Easy | 94.86 | 84.16 | **95.11** |
| | Medium | 66.27 | 61.69 | **66.97** |
| | Hard | 36.53 | 30.56 | **44.12** |
| | Overall | 69.18 | 62.65 | **70.94** |
| ChatUniVi | Easy | **59.22** | 49.04 | 55.99 |
| | Medium | **36.95** | 28.55 | 35.63 |
| | Hard | 32.21 | 18.71 | **39.07** |
| | Overall | **42.35** | 32.58 | 41.89 |
| LLaVA-NeXT | Easy | **94.24** | 91.43 | 92.44 |
| | Medium | **67.02** | 56.00 | 59.39 |
| | Hard | 33.44 | **34.21** | 30.78 |
| | Overall | **68.89** | 62.16 | 63.72 |
| mPLUG-Owl3 | Easy | 87.28 | **88.51** | 87.11 |
| | Medium | 55.58 | **58.75** | 57.37 |
| | Hard | 25.40 | 24.88 | **28.89** |
| | Overall | 59.26 | **61.27** | 60.80 |
| ST-LLM | Easy | 59.26 | **68.54** | 63.09 |
| | Medium | 36.08 | 33.23 | **41.14** |
| | Hard | 30.51 | 22.10 | **36.08** |
| | Overall | 41.59 | 40.09 | **46.39** |
| Tarsier | Easy | **89.71** | 84.01 | 88.13 |
| | Medium | **58.43** | 50.94 | 58.35 |
| | Hard | 24.25 | 24.80 | **25.17** |
| | Overall | **61.32** | 55.70 | 60.99 |
| Video-ChatGPT | Easy | 62.36 | **63.91** | 37.04 |
| | Medium | **39.40** | 38.64 | 36.01 |
| | Hard | **32.55** | 19.12 | 22.82 |
| | Overall | **44.63** | 42.36 | 34.05 |
| VideoChat2 | Easy | 88.30 | 88.62 | **89.37** |
| | Medium | 58.73 | **60.45** | 58.65 |
| | Hard | 26.94 | **28.13** | 25.29 |
| | Overall | 61.55 | **62.79** | 61.53 |

## H  EXAMPLES OF ERRORS ACROSS PROMPT STRATEGIES

In this section, we show examples where the question was answered correctly in the $X \rightarrow Y$ prompt setting but failed in either the $X \rightarrow RY$, $X \rightarrow YR$, or both settings.

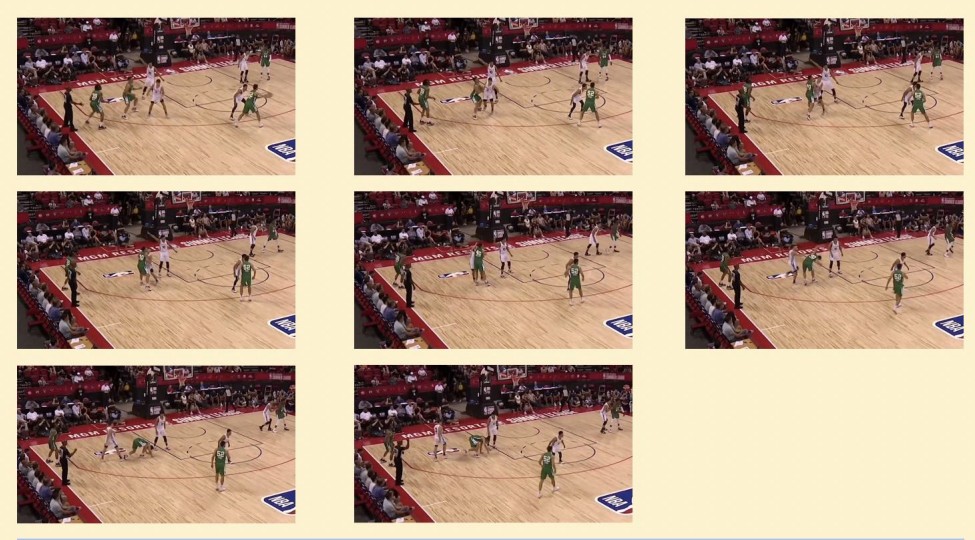

**Question:** What main color jersey does the player who committed the foul wear?
A: white&red   B: green   C: white   D: black&yellow
Ground Truth: C

$X \rightarrow Y$: C

$X \rightarrow RY$: To determine the main color jersey of the player who committed the foul, let's analyze the video frames step by step: 1. The video shows a basketball game in progress.\n2. There are two teams on the court: one wearing white jerseys and the other wearing green jerseys.\n3. In the later frames, we can see a player in a green jersey making contact with a player in a white jersey near the center of the court.\n4. The referee, visible in black, is seen raising his arm, which typically indicates a foul call.\n5. The player in the green jersey is the one who appears to be committing the action that leads to the foul call. Based on these observations, we can conclude that the player who committed the foul is wearing a green jersey. Therefore, the correct answer is: B The correct answer is: B

$X \rightarrow YR$: The correct answer is: B I chose this option because in the video frames, we can clearly see that the player who commits the foul is wearing a green jersey. The sequence of images shows a player in a green uniform making contact with a player in a white uniform, which appears to be the foul in question. The other players on the same team are also wearing green jerseys, confirming that this is one of the team colors. None of the other color options (white&red, white, or black&yellow) match the jersey color of the player committing the foul."

Figure 6: Basketball Easy level Question

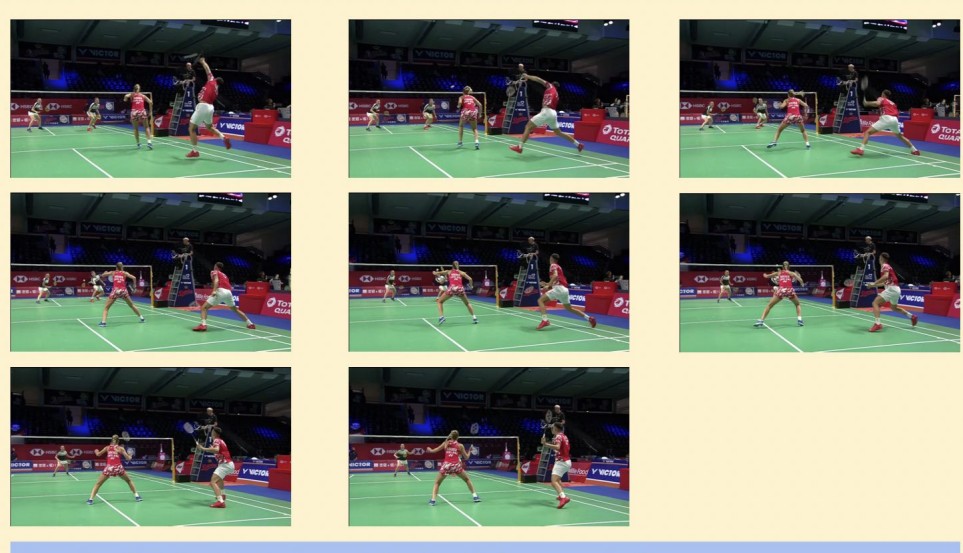

**Question:** What main color jersey is the far side team wearing?
A: Green B: Purple C: Red  D: White
Ground Truth: D

X → Y: D

X → RY: To determine the main color jersey of the far side team, I'll analyze the video frames step by step: 1. Identify the players: There are two players visible in most frames. 2. Determine which player is on the far side: The player wearing a darker jersey is consistently positioned farther from the camera, making them the \"far side\" player. 3. Observe the jersey color of the far side player: The player on the far side is wearing a dark-colored jersey. 4. Analyze the color options provided: The options are Green, Purple, Red, and White. 5. Compare the observed color to the options: The dark jersey appears to be black or a very dark shade. Among the given options, the closest match would be option B: Purple, as it's the darkest color provided. 6. Confirm consistency: Throughout all frames, the far side player's jersey remains consistently dark. 7. Consider lighting and video quality: While the jersey appears black, the lighting and video quality might affect color perception. Purple is the most plausible option among those given. Based on this analysis, the main color jersey of the far side team is most likely to be Purple, as it's the darkest color option provided and the closest match to the observed jersey color in the video frames. The correct answer is: B

X → YR: The correct answer is: D The far side team is wearing a white jersey. In the video frames, we can clearly see one player wearing a black jersey in the foreground, who appears to be from China based on the text on their jersey. The opponent, who is on the far side of the court, is wearing a predominantly white jersey with some pink or red accents. This white and pink/red jersey is consistent across all the frames where the player is visible.

Figure 7: The same question is answered correctly in the $X \rightarrow Y$ setting but incorrectly in both the $X \rightarrow RY$ and $X \rightarrow YR$ settings.

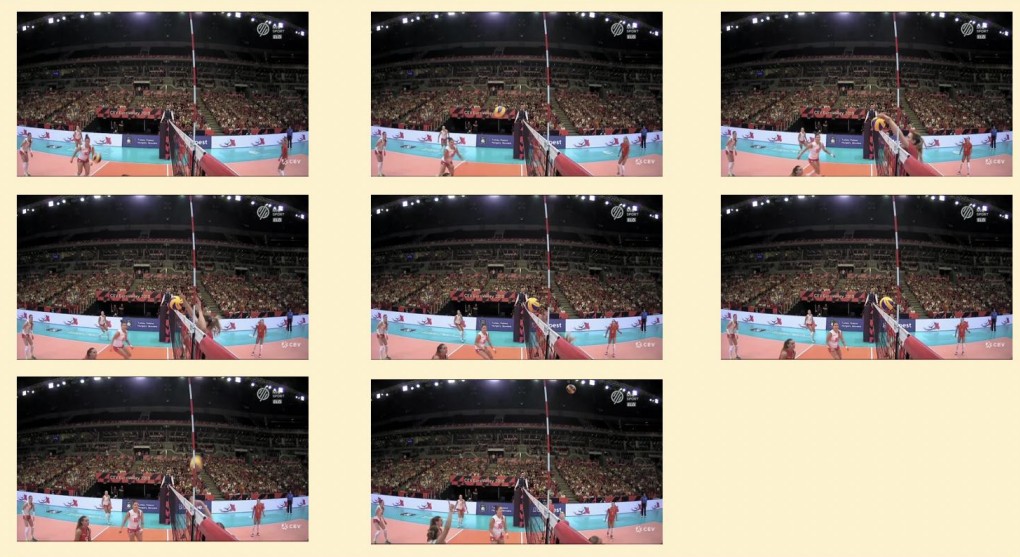

**Question:** What kind of fault or foul does this video show?
A: cross the center line  B: double hit C: over reach D: ilegal attack
Ground Truth: B

$X \to Y$: B

$X \to RY$: Based on the video frames provided, I can analyze the situation step by step:1. The images show a volleyball match at the CEV EuroVolley 2019 tournament.2. We can see players from two teams on either side of the net.3. In the later frames, we see a player reaching over the net to contact the ball.4. This action is known as  reaching over the net  or over-reaching  in volleyball.5. Reaching over the net to interfere with the opponent's play is generally not allowed in volleyball, except under specific circumstances.6. This action doesn't appear to be a center line violation, a double hit, or an illegal attack based on what we can see in the images.Given these observations, the most likely fault or foul shown in this video is over reach  or reaching over the net.The correct answer is: C"

$X \to YR$: The correct answer is: C\ I chose option C,  over reach,  because the video frames show a volleyball player reaching over the net to hit the ball. In volleyball, reaching over the net to play the ball on the opponent's side is a fault known as  over reaching  or  reaching beyond the net. In the sequence of images, we can see a player's hand and arm extending above and beyond the top of the net as they attempt to play the ball. This action is not allowed in volleyball as players must respect the opponent's playing space. The ball must be played on your own side of the net, except in specific situations like blocking.The other options don't fit what we see in the images:A)  Cross the center line  isn't visible in these frames.B)  Double hit isn't apparent from a single contact shown.D)  Illegal attack  is too vague and doesn't specifically describe the over-reaching we can see.Therefore, based on the visual evidence in the video frames, the fault being committed is most likely an  over reach,  making option C the correct answer.

Figure 8: The same question is answered correctly in both the $X \to Y$ setting and $X \to YR$ settings but incorrectly in the $X \to RY$ setting.

# I CORRELATION MATRIX OF EVALUATION METRICS

We applied Pearson correlation to assess the relationship between the automatic evaluation metrics and human-annotated scores for the SPORTU-Video part. As shown in Figure 9, G-eval has the highest correlation with the human scores among all the automatic metrics. However, the correlation is still weak, indicating the need for the development of a new evaluation metric in the future to improve the assessment process.

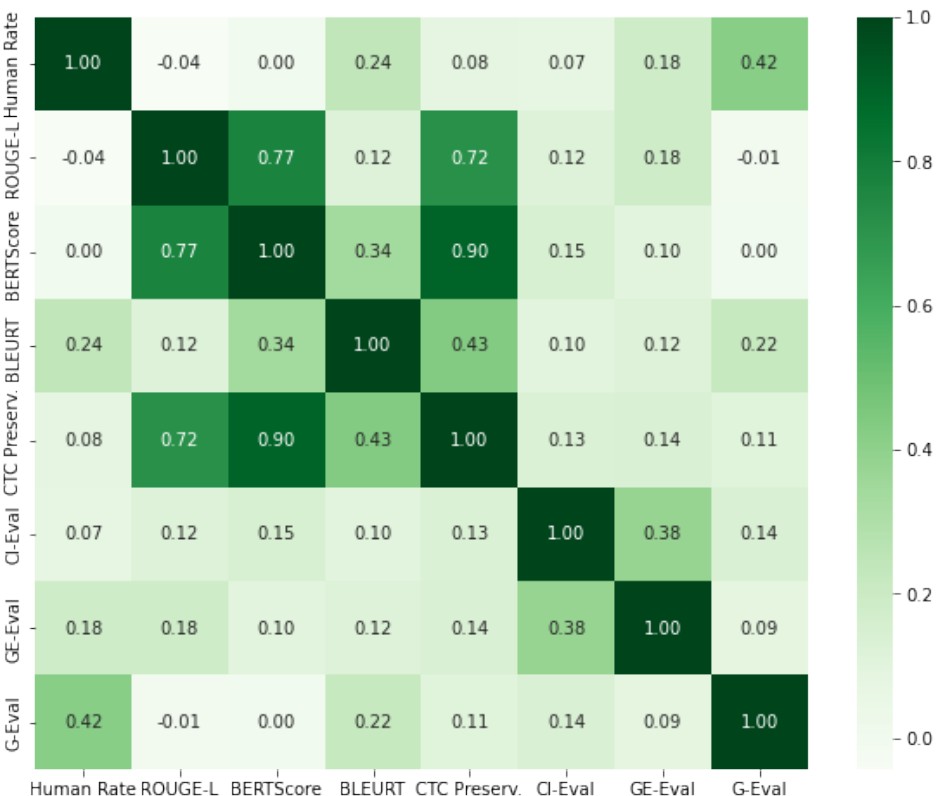

Figure 9: Examples of SPORTU-video

# J CRITEIRA OF G-EVAL

## CRITERIA OF G-EVAL

This score assesses the overall accuracy, conciseness, and relevance of the model-generated explanation to the ground truth explanation. The explanation should focus on identifying and clearly explaining the key issue or relevant action.

Scoring Breakdown:

- Score 1 (very poor):
  The explanation is incorrect or does not address the key issue at all. It might be filled with irrelevant details that distract from the main point.
- Score 2 (poor):
  The explanation contains some correct elements but fails to clearly identify the key issue. There may be excessive irrelevant details that obscure the main point.
- Score 3 (adequate):
  The explanation identifies the key issue but includes too much unnecessary information or lacks clarity. The key point is present but could be more concise.
- Score 4 (good):
  The explanation clearly identifies the key issue with some minor unnecessary details. The core explanation is correct and relevant but may include a few extra, non-essential details.
- Score 5 (excellent):
  The explanation is concise, accurate, and directly addresses the key issue without any unnecessary information. It clearly and effectively answers the question.

Evaluation Steps:

1. Read the ground truth explanation.
2. Identify the specific context in the ground truth explanation that represents the key issue (e.g., a foul, a mistake).
3. Read the model output explanation.
4. Compare the model's explanation to the ground truth explanation, focusing on how directly and concisely the model output identifies the key issue.
5. Assign a score for explanation of overall relevance on a scale of 1 to 5, based on the criteria.

Model Generated Explanation: {{Model Generated Content}}
Ground Truth Explanation: {{Ground Truth Content}}
Evaluation Form (scores ONLY): Overall:

Figure 10: Criteria and prompt used in G-Eval score evaluation. The same prompt template is applied to all four evaluator models.

# K SPORTU-TEXT PROMPT TEMPLATES

## K.1 ZERO SHOT STANDARD PROMPT

Table 12: Prompt Template for Zero-shot Standard Prompt on SPORTU-text

```
{
  role: "system",
  content: "You are a sport experts answering sports-related questions. Please indicate
the correct answer(s) clearly with letter(s)."
},
{
  role:"user",
  content: "Question: {{question}} {{options}}. Only output the correct option letters."
}
```

## K.2 ZERO SHOT CoT PROMPT

Table 13: Prompt Template for Zero-shot CoT Prompt on SPORTU-text

```
{
  role: "system",
  content: "You are a sport experts answering sports-related questions. Please indicate
the correct answer(s) clearly with letter(s)."
},
{
  role:"user",
  content: " Question: {{question}} {{options}}. Let's think step by step."
}
```

## K.3 FIVE SHOT STANDARD PROMPT

Table 14: Prompt Template for Five-shot Standard Prompt on SPORTU-text

```
{
    role: "system",
    content: "You are a sport experts answering sports-related questions. Please indicate
the correct answer(s) clearly with letter(s)."
},
{{five-shot examples}}, {
    role:"user",
    content: "Question: {{question}} {{options}}. Only output the correct option letters."
}
```

## K.4 FIVE SHOT COT PROMPT

Table 15: Prompt Template for Five-shot CoT Prompt on SPORTU-text

```
{
    role: "system",
    content: "You are a sport experts answering sports-related questions. Please indicate
the correct answer(s) clearly with letter(s)."
},
{{five-shot examples with explanations}}, {
    role:"user",
    content: "Question: {{question}} Options: {{options}}."
}
```

## L  SPORTU-VIDEO PROMPT TEMPLATES

### L.1  X - YR PROMPT TEMPLATE

Table 16: Prompt Template for SPORTU-video $X \rightarrow YR$ setting

{
  role: "system",
  content: "You are a sports expert analyzing a series of video frames that form a continuous short video clip. Your task is to answer questions based on the content of these frames."
},
{
  role: "user",
  content: [
    {{video_frames}},
    {
      type: "text",
      text: "Based on the video frames provided, answer this sports-related question: {{question}} Options: {{options}}. Respond with the letter of the correct option, formatted as 'The correct answer is: ', and explain why you chose that option."
    }
  ]
}

### L.2  X - RY PROMPT TEMPLATE

Table 17: Prompt Template for SPORTU-video $X \rightarrow RY$ setting

{
  role:"system",
  content: "You are a sports expert analyzing a series of video frames that form a continuous short video clip. Your task is to answer questions based on the content of these frames."
},
{
  role: "user"
  content: [
    {{video_frames}},
    {
      type: "text",
      text: "Based on the video frames provided, answer this sports-related question: {{question}} Options: {{options}}. Let's answer this question step by step. add a sentence formatted as 'The correct answer is: ' at the end of your thinking process. "
    }
  ]
}

## L.3    X - Y PROMPT TEMPLATE

Table 18: Prompt Template for SPORTU-video $X \rightarrow Y$ setting

```
{
  role: "system",
  content: "You are a sports expert analyzing a series of video frames that form a con-
tinuous short video clip. Your task is to answer questions based on the content of these
frames."
},
{
  role: "user",
  content: [
    {{video_frames}},
    {
      type: "text",
      text: "Based on the video frames provided, answer this sports-related question:
{{question}} Options: {{options}}. Answer with only the letter of the correct option."
    }
  ]
}
```

## L.4    OPEN-ENDED TEMPLATE

Table 19: Prompt Template for SPORTU-video Open-ended Questions

```
{
  role: "system",
  content: "You are a sports expert analyzing a series of video frames that form a con-
tinuous short video clip. Your task is to answer questions based on the content of these
frames."
},
{
  role: "user",
  content: [
    {{video_frames}},
    {
      type: "text",
      text: "Based on the video frames provided, answer this sports-related question:
{{question}}. Please provide a detailed explanation, focusing on the sports aspects of the
video."
    }
  ]
}
```

# M   ADDITIONAL ERROR ANALYSIS

This section presents the error analysis of the models we evaluated on SPORTU-video. For the error types, we evaluated the models with coarse granularity, dividing the errors into five categories as follows:

- Question Understanding Error – The model misinterprets the intent or context of the question, providing an answer that does not align with what the question is asking.

- Visual Perception Error – The model incorrectly interprets the visual content, leading to faulty assumptions about the data presented in the video.

- Hallucinations – The model generates content or details that do not exist in the actual data, essentially 'hallucinating' information.

- Reasoning Error – The model exhibits poor logical reasoning, resulting in incorrect conclusions based on the available data.

- Lack of Domain Knowledge – The model fails to answer questions that require specific domain expertise, revealing a gap in its knowledge.

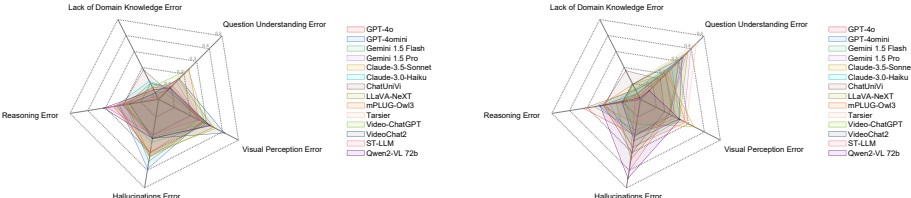

Figure 11: Error type distribution across different MLLMs on SPORTU-video tasks: the left side represents errors from multiple-choice questions, while the right side represents errors from open-ended questions.

We observe that open-ended questions have the highest frequency of question understanding errors. For instance, when asked 'Why is it a foul in the video?', the model might respond that there is no foul, which is a question understanding error because the phrasing of the question already implies that a foul occurred. This issue is much less frequent in multiple-choice questions (as shown on the left side of the figure 11). When presented with four options, the model tends to select one of the provided answers, rather than completely misinterpreting the premise of the question. Examples of each Error are in Appendix Q.

# N QUESTION TEMPLATE FOR EACH SPORT

This section shows the questions we use for each sport. (Table 20 - Table26).

Table 20: Question Template for Volleyball

| |
|---|
| Why is it a fault in the video? |
| What sport does this video show? |
| What kind of fault or foul does this video show? |
| What main color jersey is the libero wearing? |
| If the libero's jersey color does not count, what main color jerseys are the players on the right side wearing? |
| If the libero's jersey color does not count, what main color jerseys are the players on the left side wearing? |
| If the libero's jersey color does not count, what main color jerseys are the players on the far side wearing? |
| If the libero's jersey color does not count, what main color jerseys are the players on the close side wearing? |
| What main color jersey is the libero on the close side wearing? |
| What main color jersey is the libero on the far side wearing? |
| What main color jersey is the libero on the left side wearing? |
| What main color jersey is the libero on the right side wearing? |
| Is there a rule violation in the video? |

Table 21: Question Template for Basketball

| |
|---|
| What sport does this video show? |
| What specific type of foul, if any, occurred in the video? Choose the most appropriate one. |
| What main color jersey is the offensive team wearing? |
| What main color jersey is the defensive team wearing? |
| What main color jersey does the player who committed the foul wear? |
| What main color jersey is the player who was fouled wearing? |
| What main jersey colors do the two teams wear in this video? |
| Is there a rule violation in the video? |
| Why is it a foul in the video? |

Table 22: Question Template for Badminton

| |
|---|
| Why is it a rule violation in the video? |
| How is the player making a technical error in the video? |
| What sport does this video show? |
| What kind of rule violation is in the video? |
| What main color jersey is the left side team wearing? |
| What main color jersey is the right side team wearing? |
| What main color jersey is the close side team wearing? |
| What main color jersey is the far side team wearing? |
| What specific action does the player perform in the video? |
| How many players are shown in total in this video? |
| Is the player making a technical error in the video? |
| Is there a rule violation in the video? |

Table 23: Question Template for Baseball

| |
|---|
| What sport does this video show? |
| What kind of rule violation is in the video? |
| Based on this video, which of the following descriptions best applies to the situation? |
| As a referee, what procedure would you follow in this situation in the video? |
| What main color jersey is the fielder team wearing? |
| What main color jersey is the batting team wearing? |
| What main jersey colors do the two teams wear in this video? |
| Is there a rule violation in the video? |
| Why did the rule violation occur in the video? |

Table 24: Question Template for Soccer

| |
|---|
| Why is it a foul in the video? |
| Why is it not a foul in the video? |
| Based on this video, which of the following descriptions apply to the situation that occurred? |
| What sport does this video show? |
| What kind of foul does this video show? |
| What main color jersey is the offensive team wearing? |
| What main color jersey is the defensive team wearing? |
| What main color jersey does the player who committed the foul wear? |
| What main color jersey is the player who was fouled wearing? |
| What main color jersey does the goalkeeper wear? |
| What main jersey colors do the two teams wear in this video? |
| How many players are shown in total in this video? |
| Is there a rule violation in the video? |

Table 25: Question Template for Ice Hockey

| |
|---|
| What sport does this video show? |
| What kind of foul is committed in the video? |
| What main color jersey does the player who committed the foul wear? |
| What main color jersey is the player who was fouled wearing? |
| What main jersey colors do the two teams wear in this video? |
| Is there a rule violation in the video? |
| If we consider the fight in the video to be a legit fight, defined as a fight between two willing participants who drop their gloves and helmets, with the fight ending when one player falls or officials intervene, and this fight does not result in a penalty, is there any other foul in the video that will cause a penalty? |
| Why is it a foul in the video? |

Table 26: Question Template for American Football

| |
|---|
| Why is it an error in the video? |
| Why is it a foul in the video? |
| What sport does this video show? |
| What kind of foul does this video show? |
| What kind of error does the player make in this video? |
| What main color jersey is the offensive team wearing? |
| What main color jersey is the defensive team wearing? |
| What main jersey colors do the two teams wear in this video? |
| Is there a foul in this video? |
| Does any player make an error in this video? |

## O  SPORTU-TEXT EXAMPLES

### O.1  RULE-RELATED QUESTION

Table 27: Rule Question 1

**Question**: How does ball possession work in basketball after the successful final free throw?
A) The team that made the free throw retains possession.
B) The team that missed the free throw gains possession.
C) The opposing team gains possession.
D) The team with the most points gains possession.
**Answer**: C
**Explanation**: According to FIBA's rules on what should happen after a successful field goal or the last successful free throw, following the last successful free throw, any player from the non-scoring team should inbound the ball from any position behind their own endline; therefore, option A is incorrect. The team that missed the free throw is the same team as the one executing the free throw during the game, but the question specifies that the team's last free throw was successful; therefore, option B is incorrect. Since the rule states that any player from the non-scoring team should inbound the ball from behind their own endline, and since the team that executed the last free throw scored, the non-scoring team, i.e., the opposing team, should regain possession of the ball and inbound it; therefore, option C is correct. The scoring situation of both teams does not affect the distribution of possession after a successful free throw, so neither the number of points scored nor the point differential is a determining factor for possession distribution; therefore, option D is incorrect. Hence, the correct answer is option C.

Table 28: Rule Question 2

**Question**: Question: Which player is typically responsible for throwing the ball to the receivers in American football?
A) Linebacker.
B) Quarterback.
C) Running back.
D) Tight end.
**Answer**: B
**Explanation**: Linebacker is on the defensive side and does not often have a chance to touch the ball, therefore option A is incorrect. Quarterback usually throws the ball to running back, therefore option B is correct. Running back is the one who often receives the ball, therefore option C is incorrect. Tight end usually blocks the other player and is not primarily meant to grab the ball, therefore option D is incorrect. Hence, the correct answer is option B.

## O.2 STRATEGY-RELATED QUESTION

Table 29: Strategy-related Question Sample 1

| |
|---|
| **Question**: What are key considerations when implementing a successful blocking scheme in volleyball defensive strategies?
A) The blocker's positioning in relation to the attacker's hitting arm.
B) The speed of the incoming serve.
C) The position of the setter on the opposing team.
D) The timing and coordination between the blockers.
**Answer**: ABCD
**Explanation**: The positioning in relation to the attacker's hitting arm is crucial for a successful block and for the entire team's block and defensive formation. The side blocker usually positions their head in line with the hitter's arm and the ball to cover the straight line, or uses their right hand on the hitter's arm and the ball to cover the cross-court angle. The back-row players will adjust their position based on the main angles that the blockers are covering. Therefore, option A is correct. Increasing the speed of the serve makes it harder for the opponent to receive, potentially putting them out of system. This reduces the opponent's attacking options compared to when they are in system, making blocking easier. Therefore, option B is correct. The setter's position directly determines the team's strategic options and makes it easier for blockers to read and predict the play. Therefore, option C is correct. Timing and coordination between the blockers are also essential, as blockers aim to jump together without leaving gaps if they decide to execute a multi-blocker jump. Therefore, option D is correct. Hence, the correct answer to this question is A, B, C, and D. |

Table 30: Strategy-related Question Sample 2

| |
|---|
| **Question**: Question: Which of the following are common attack patterns in volleyball offensive strategies?
A) Quick set
B) Slide attack
C) 5-1 formation
D) Triple quick
**Answer**: AB
**Explanation**: A quick set is an attacking pattern for the middle blocker. The middle blocker jumps close to the setter and jumps before the setter sets the ball. The setter then sets the ball low and quickly to the middle blocker's optimal attacking position. Therefore, option A is correct. A slide attack is another attacking pattern for the middle blocker, where the middle blocker approaches the right-side position, usually 2-3 meters from the setter, and swings with a one-foot jump. The setter will set a low arc ball backward to the right side. Therefore, option B is correct. The 5-1 formation is a team strategy concerning how many setters are on the court and is not related to attack patterns. Therefore, option C is incorrect. `Triple quick`is not a common attack pattern or strategy in standard volleyball. Therefore, option D is incorrect. Hence, the correct answers are A and B. |

## O.3 SCENARIO-RELATED QUESTION

Table 31: Scenario-Related Question 1

| |
|---|
| **Question**: Which of the following scenarios would most likely result in a red card offense in a football match? 
 A) A player politely disagrees with the referee's decision. 
 B) A player accidentally trips another player while trying to get the ball. 
 C) A player uses offensive language or gestures towards the referee. 
 D) A player passes the ball back to his own goalkeeper. 
 **Answer**: C 
 **Explanation**: Disagreeing with the referee's decision, as long as it is done reasonably and without abuse, will not result in a red card, therefore option A is incorrect. Accidentally tripping a player will not result in a red card because it is not intentional, therefore option B is incorrect. Using offensive language or gestures violates sportsmanship and will result in a red card, therefore option C is correct. Passing the ball back to the goalkeeper does not result in a red card; it will result in an indirect free kick, therefore option D is incorrect. Hence, the correct answer is option C. |

Table 32: Scenario-Related Question 2

| |
|---|
| **Question**: In a competitive basketball game, Player A accidentally knocks the ball which then bounces off Player B's hand, rolls on the boundary line, touches Player C's foot while he is standing on the line, and finally goes out of the court. Which player is considered to have caused the ball to go out of bounds? 
 A) A) Player A because he accidentally knocked the ball first. 
 B) Player B because the ball touched his hand before rolling on the boundary line. 
 C) Player C because the ball touched his foot while he was standing on the boundary line. 
 D) None of the players caused the ball to go out of bounds because the ball rolled on its own. 
 **Answer**: C 
 **Explanation**: According to FIBA's rules regarding out-of-bounds, only the player who last touched or was touched by the ball before it went out of bounds is considered responsible for the out-of-bounds situation. Since Player A touched the ball before it touched Player B, Player A was not the last player to touch the ball; therefore, option A is incorrect. Similarly, after Player B touched the ball, it then touched Player C, so Player B was not the last player to touch the ball; therefore, option B is incorrect. Because Player C was the last player to touch the ball before it was deemed out of bounds, and the ball touched Player C's foot, causing it to go out of bounds, Player C is considered responsible for the out-of-bounds situation; therefore, option C is correct. Since the ball touched at least one player before going out of bounds, it cannot be considered that no player caused the ball to go out of bounds; therefore, option D is incorrect. Hence, the correct answer is option C. |

## P SROUTU-VIDEO EXAMPLES

### P.1 BASKETBALL

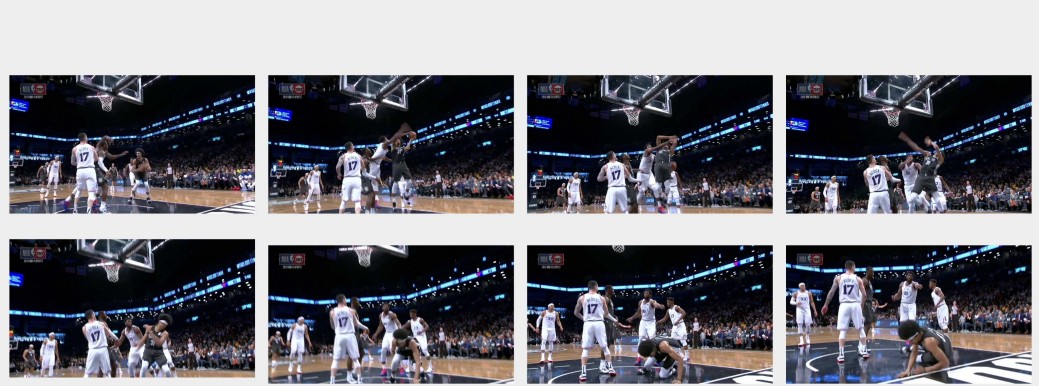

Question:In this video, what are the main jersey colors of the two teams?
A: One team wears purple jersey, the other team wears red jersey,
B: One team wears red & blue jersey, the other team wears red jersey,
C: One team wears black&orange jersey, the other team wears red & blue jersey,
D: One team wears black jersey, the other team wears white jersey
Answer: D

Figure 12: Basketball Easy level Question

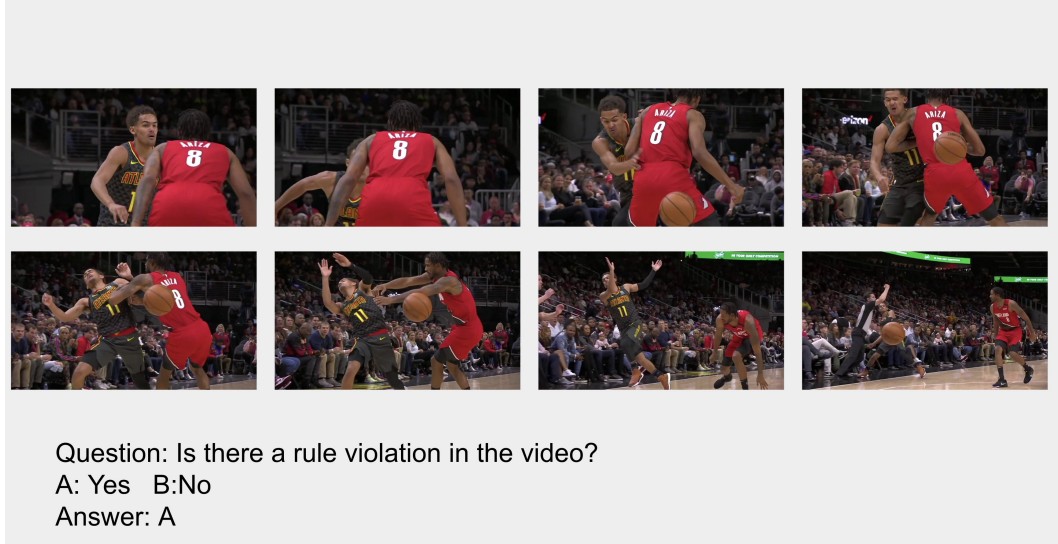

Question: Is there a rule violation in the video?
A: Yes   B:No
Answer: A

Figure 13: Basketball Medium level Question

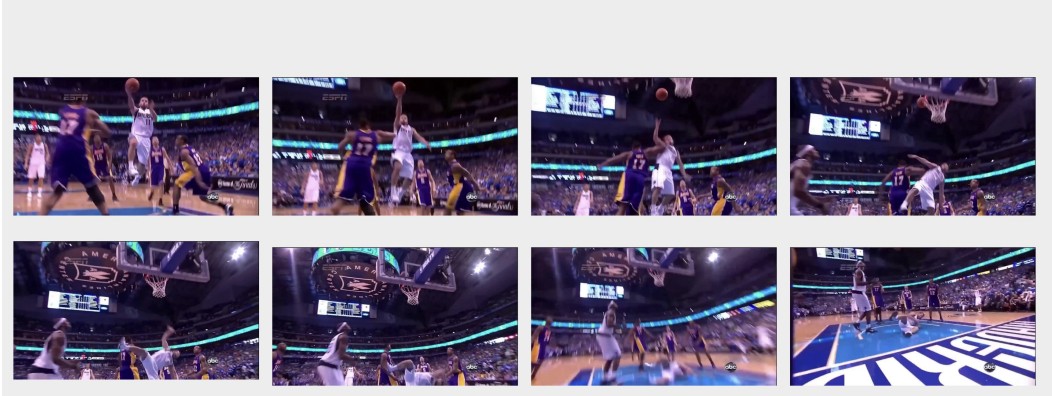

Question:Why is it a foul in the video?
Answer: The player in the purple and yellow jersey elbowed the shooting player in white in their right armpit.

Figure 14: Basketball hard level Question

## P.2 VOLLEYBALL

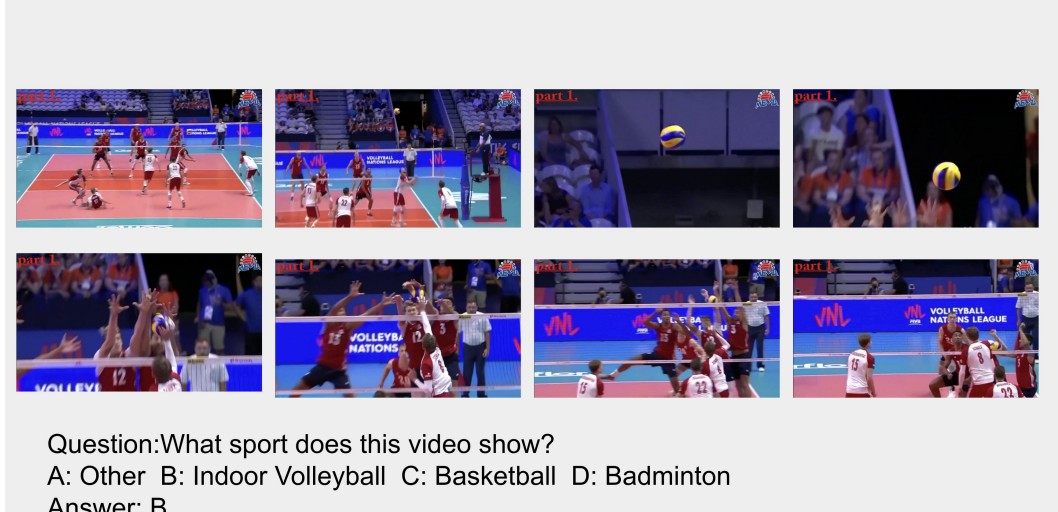

Question:What sport does this video show?
A: Other  B: Indoor Volleyball  C: Basketball  D: Badminton
Answer: B

Figure 15: Volleyball easy level Question

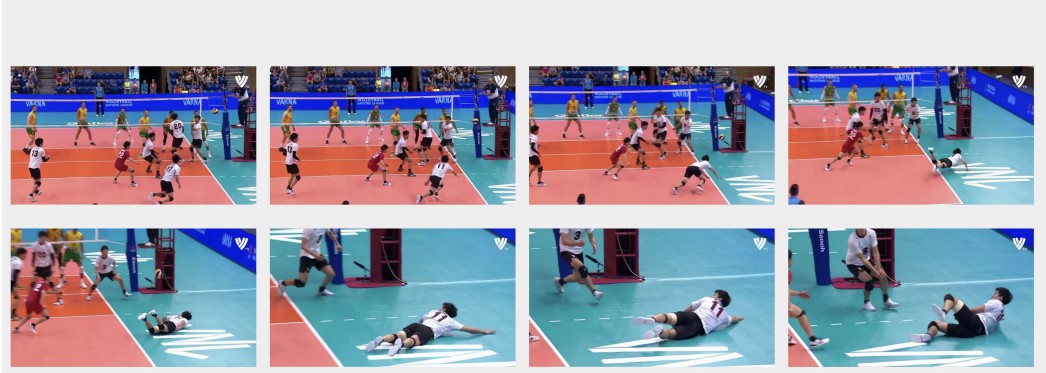

Question:If the libero's jersey color does not count, what main color jerseys are the players on the close side wearing?
A: white  B: red  C: yellow  D: white&red
Answer: A

Figure 16: Volleyball medium level Question

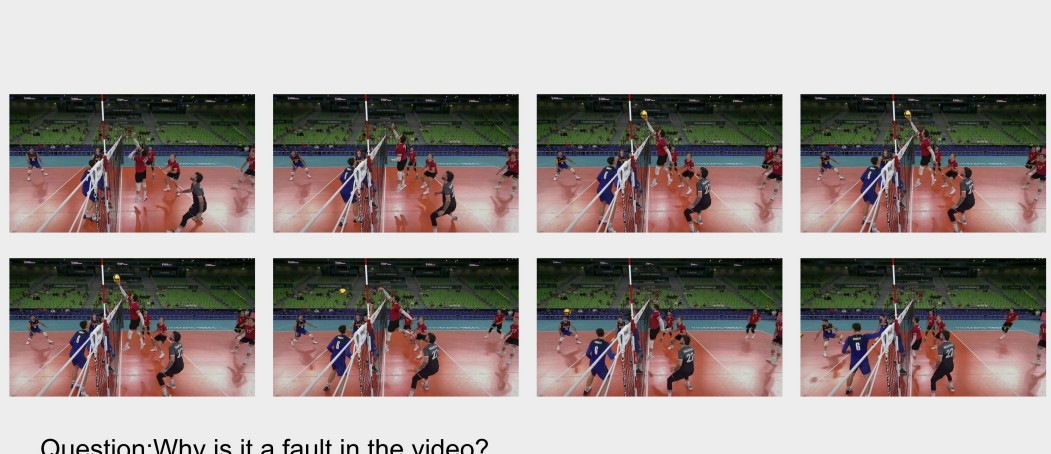

Question:Why is it a fault in the video?
Answer: The player in the red jersey on the right side reached over the net with their right hand while hitting the ball

Figure 17: Volleyball hard level Question

P.3    SOCCER

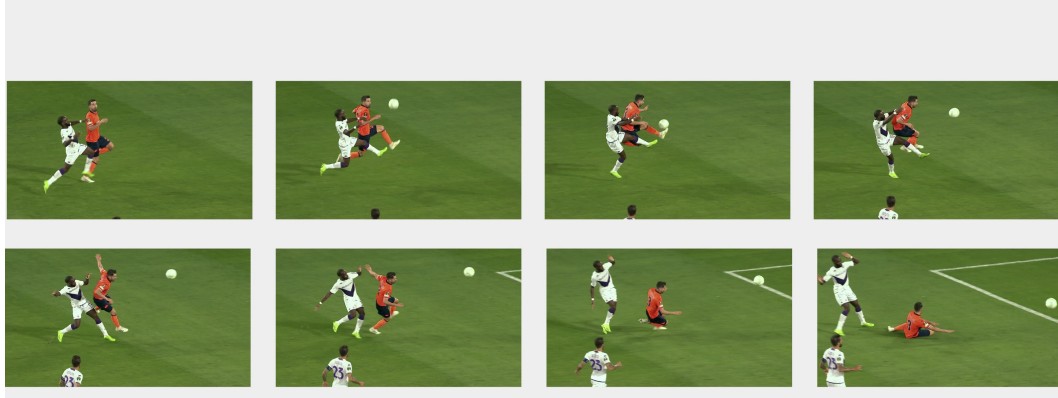

Question: How many players are shown in total in this video?,
A: 6  B: 3  C: 9  D: 4
Answer: B

Figure 18: Soccer easy level Question

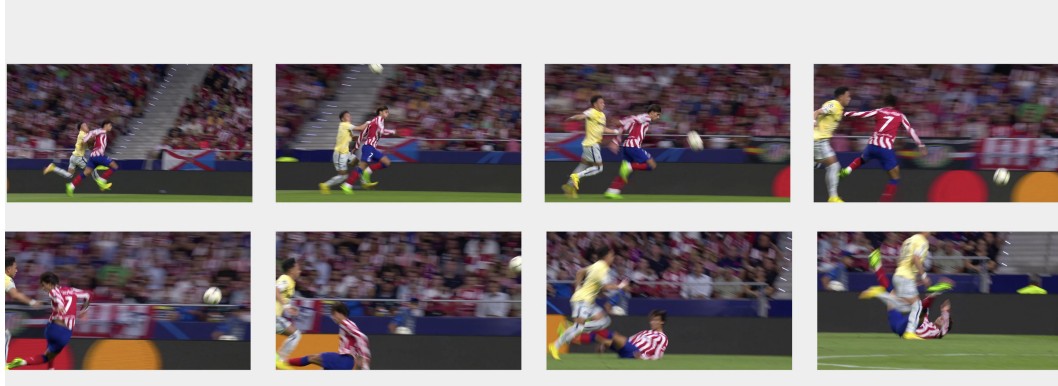

Question:Based on this video, which of the following descriptions apply to the
situation that occurred?
A: Penalty kick  B: Other  C: Offside  D: DOGSO-SPA
Answer: D

Figure 19: Soccer medium level Question

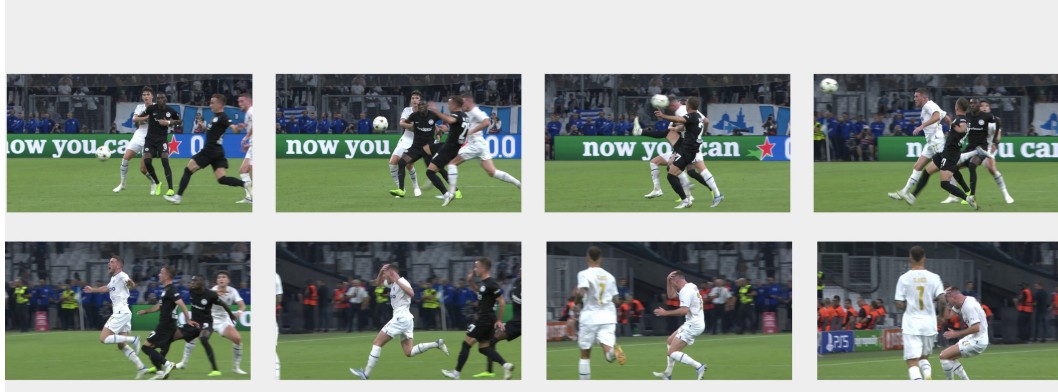

Question: What kind of foul does this video show?
A: trip foul  B: offside foul  C: pull foul D: dive foul
Answer: D

Figure 20: Soccer hard level Question

## P.4 BADMINTON

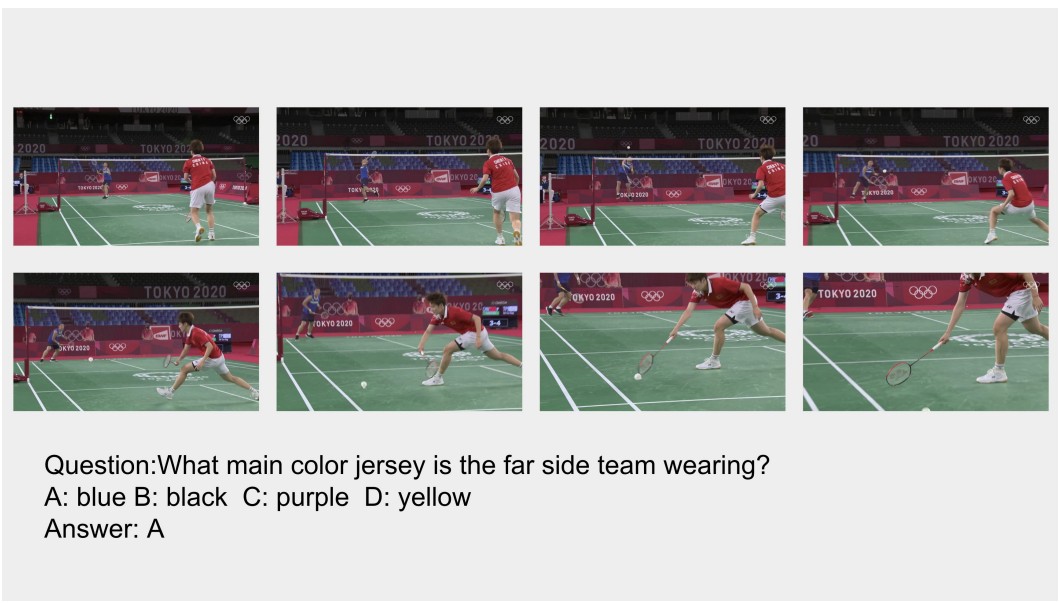

Question:What main color jersey is the far side team wearing?
A: blue B: black  C: purple  D: yellow
Answer: A

Figure 21: Badminton easy level Question

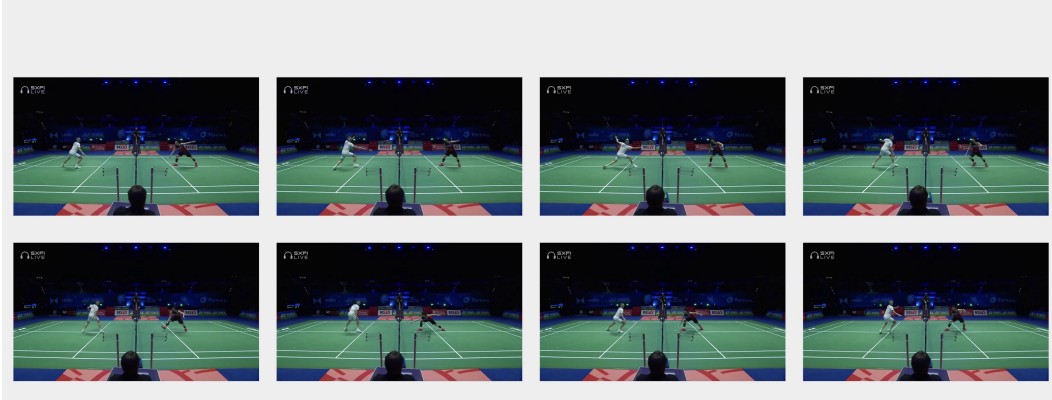

Question: What specific action does the player perform in the video?
A: Net Shot  B: Net Kill  C: Push Shot    D: Lift
Answer: A

Figure 22: Badminton medium level Question

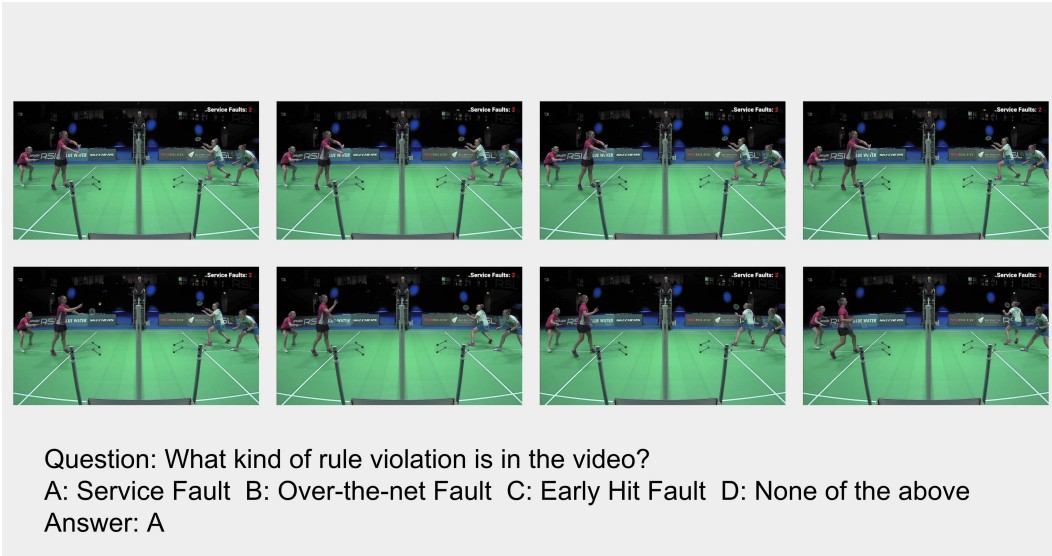

Question: What kind of rule violation is in the video?
A: Service Fault  B: Over-the-net Fault  C: Early Hit Fault  D: None of the above
Answer: A

Figure 23: Badminton hard level Question

P.5   AMERICAN FOOTBALL

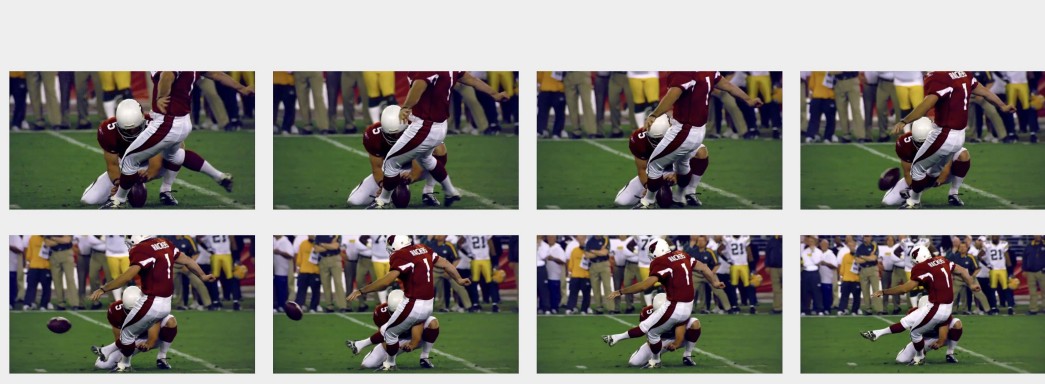

Question: What sport does this video show?
A: Baseball  B: Handball  C: Ice Hockey  D: American Football
Answer: D

Figure 24: American Football easy level Question

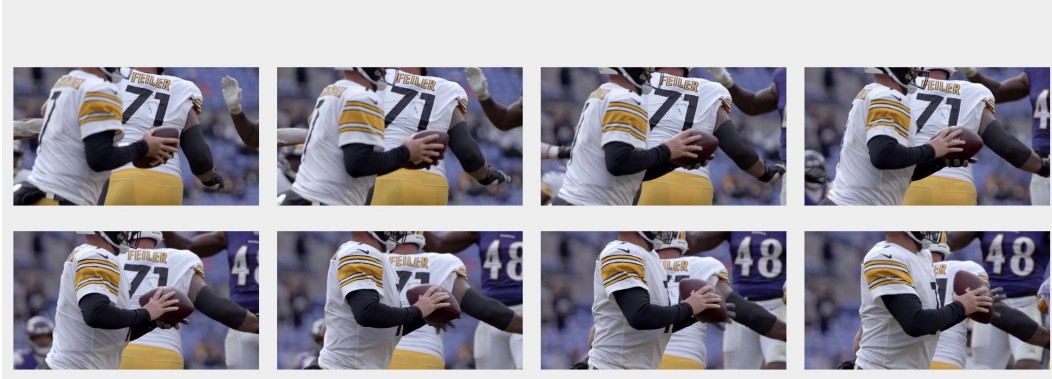

Question: What main color jersey is the offensive team wearing?
A: orange  B: blue  C: white  D: green
Answer: C

Figure 25: American Football medium level Question

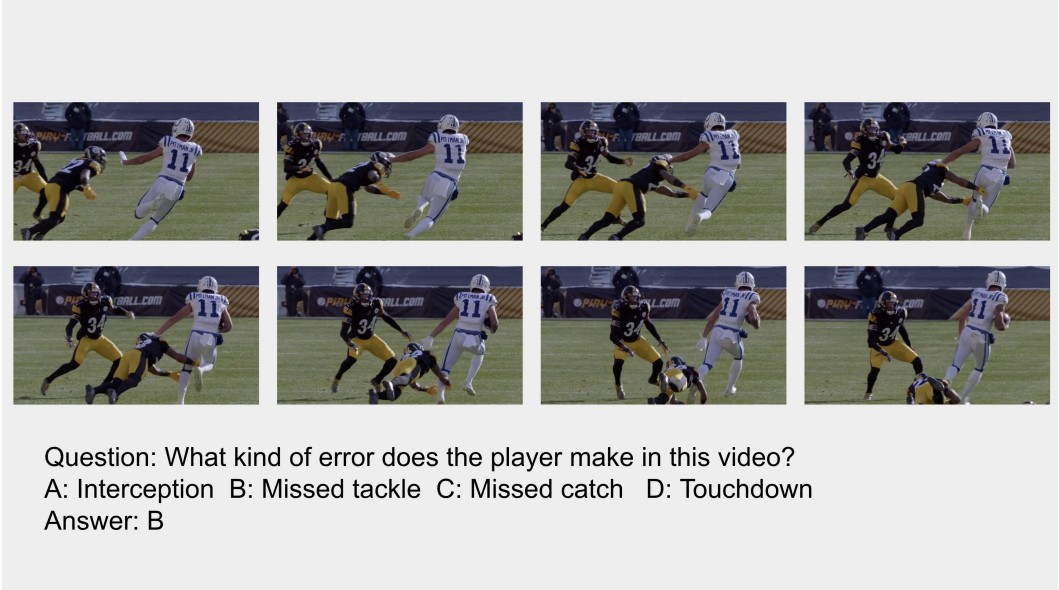

Question: What kind of error does the player make in this video?
A: Interception   B: Missed tackle   C: Missed catch   D: Touchdown
Answer: B

Figure 26: American Football hard level Question

## P.6   ICE HOCKEY

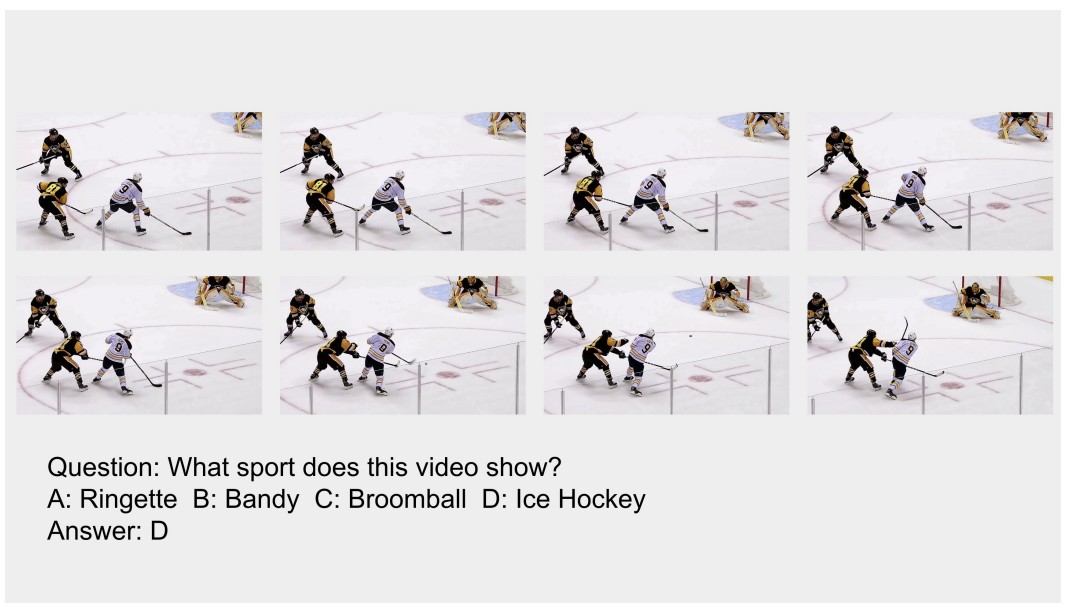

Question: What sport does this video show?
A: Ringette   B: Bandy   C: Broomball   D: Ice Hockey
Answer: D

Figure 27: Ice Hockey easy level Question

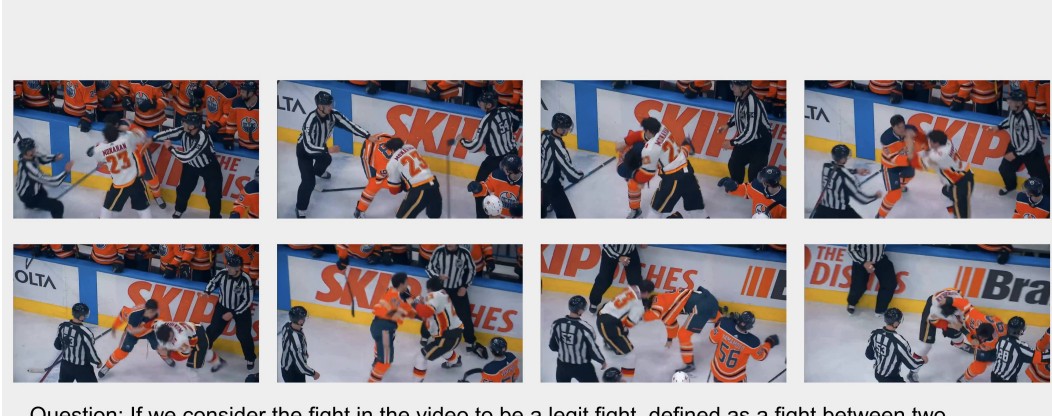

Question: If we consider the fight in the video to be a legit fight, defined as a fight between two willing participants who drop their gloves and helmets, with the fight ending when one player falls or officials intervene, and this fight does not result in a penalty, is there any other foul in the video that will cause a penalty?
A: No  B: Yes
Answer: A

Figure 28: Ice Hockey medium level Question

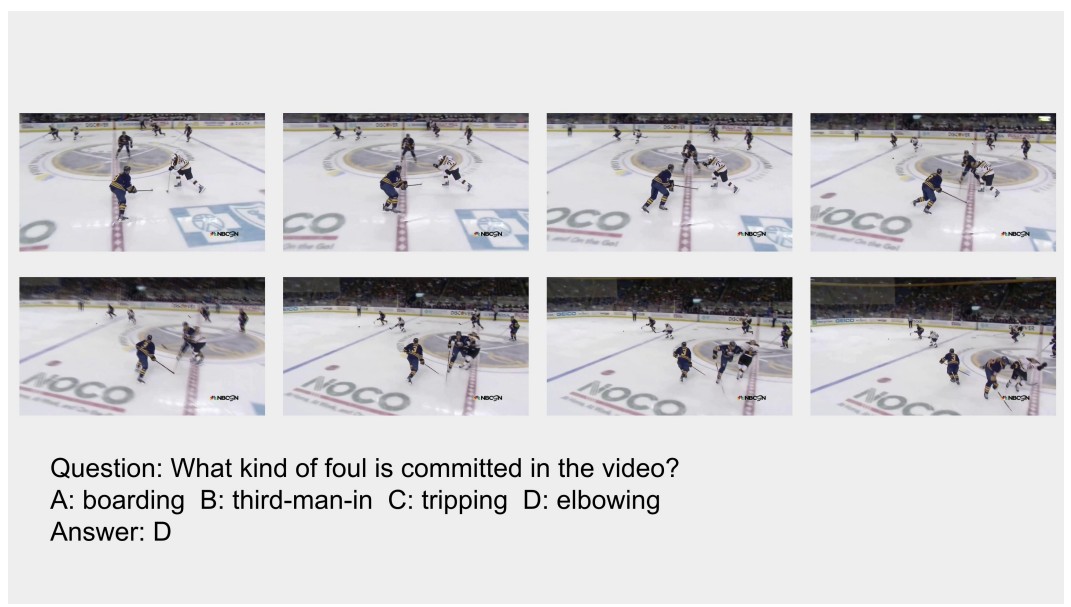

Question: What kind of foul is committed in the video?
A: boarding  B: third-man-in  C: tripping  D: elbowing
Answer: D

Figure 29: Ice Hockey hard level Question

## P.7  BASEBALL

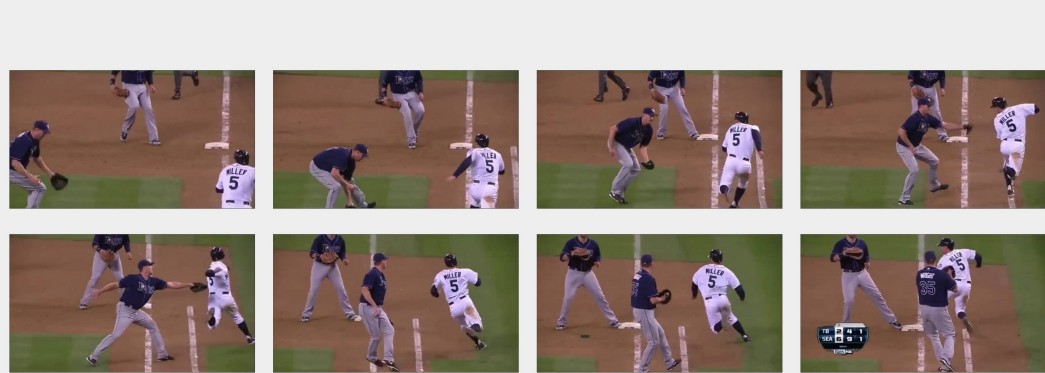

Question: In this video, what are the main jersey colors of the two teams?
A: One team wears dark blue jersey, the other team wears grey jersey
B: One team wears white&red jersey, the other team wears dark red jersey
C: One team wears dark blue jersey, the other team wears white&green jersey
D: One team wears white jersey, the other team wears dark blue jersey
Answer: D

Figure 30: Baseball easy level Question

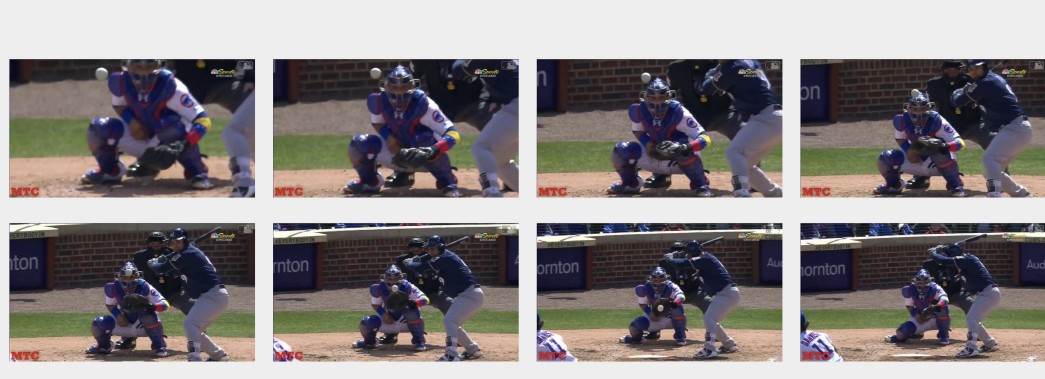

Question: What main color jersey is the batting team wearing?
A: tan  B: green  C: blue  D: gray
Answer: C

Figure 31: Baseball medium level Question

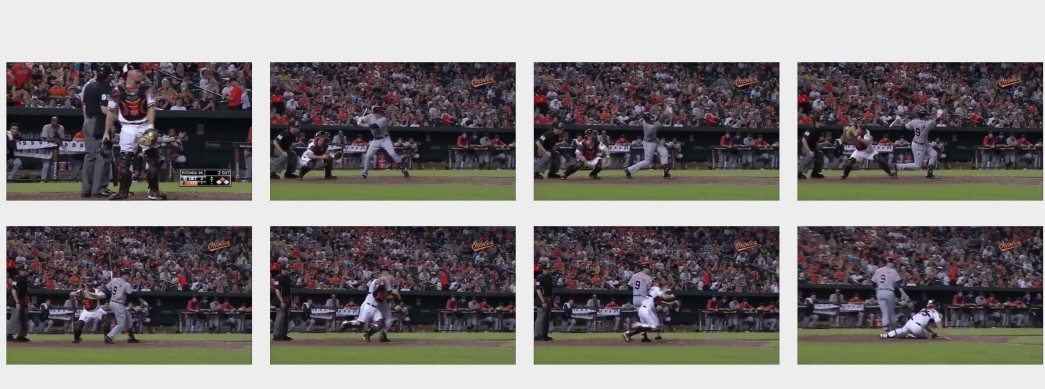

question: What kind of rule violation is in the video?
A: Balk  B: Running Out of the Baseline  C: Interference  D: None of the above
Answer: C

Figure 32: Baseball hard level Question

## Q EXAMPLES OF EACH ERROR TYPE

### Q.1 QUESTION UNDERSTANDING ERROR

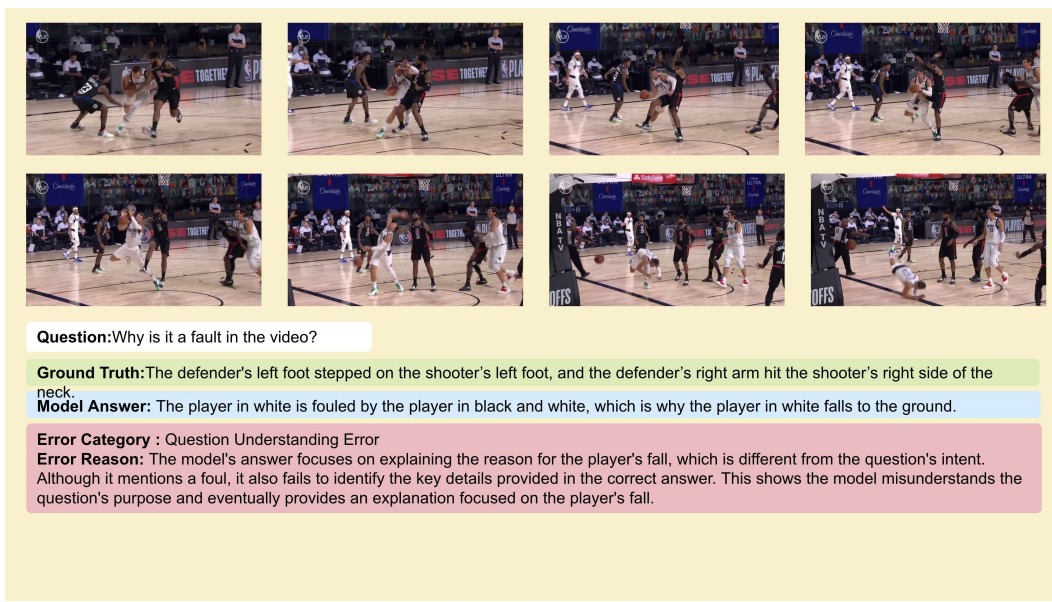

Figure 33: Examples of Question Understanding Type and Error Reason Explanations

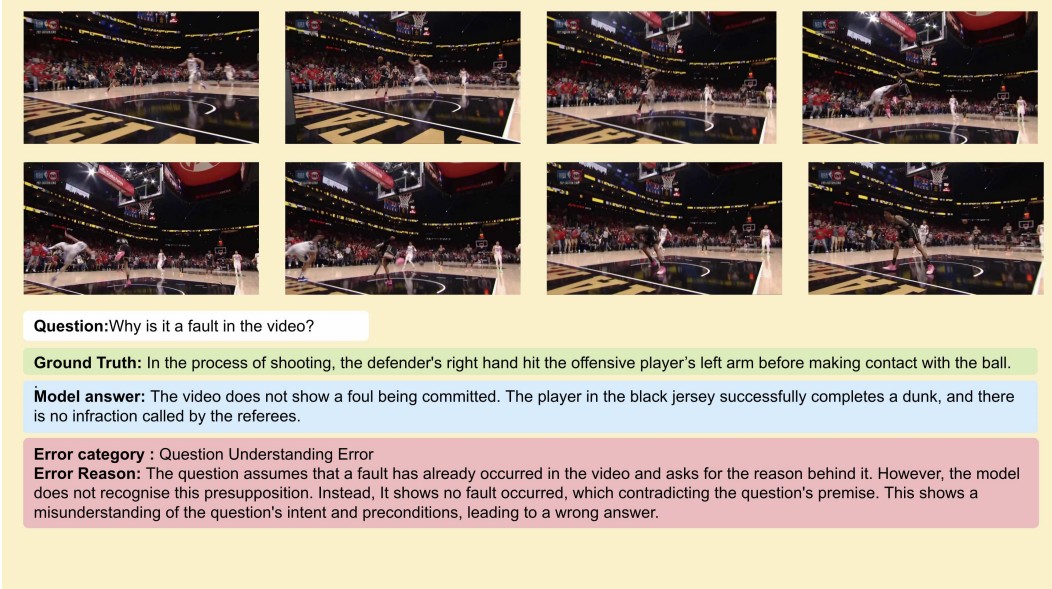

Figure 34: Examples of Question Understanding Error and Error Reason Explanation

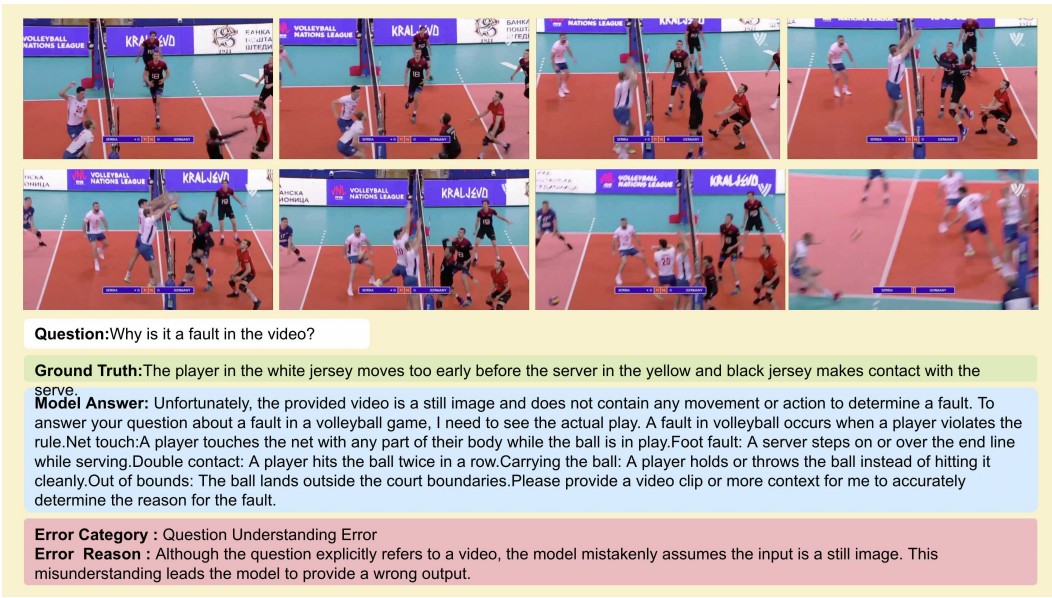

**Question:** Why is it a fault in the video?

**Ground Truth:** The player in the white jersey moves too early before the server in the yellow and black jersey makes contact with the serve.

**Model Answer:** Unfortunately, the provided video is a still image and does not contain any movement or action to determine a fault. To answer your question about a fault in a volleyball game, I need to see the actual play. A fault in volleyball occurs when a player violates the rule.Net touch:A player touches the net with any part of their body while the ball is in play.Foot fault: A server steps on or over the end line while serving.Double contact: A player hits the ball twice in a row.Carrying the ball: A player holds or throws the ball instead of hitting it cleanly.Out of bounds: The ball lands outside the court boundaries.Please provide a video clip or more context for me to accurately determine the reason for the fault.

**Error Category :** Question Understanding Error
**Error  Reason :** Although the question explicitly refers to a video, the model mistakenly assumes the input is a still image. This misunderstanding leads the model to provide a wrong output.

Figure 35: Examples of Question Understanding Error and Error Reason Explanation

## Q.2   VISUAL PERCEPTION ERROR

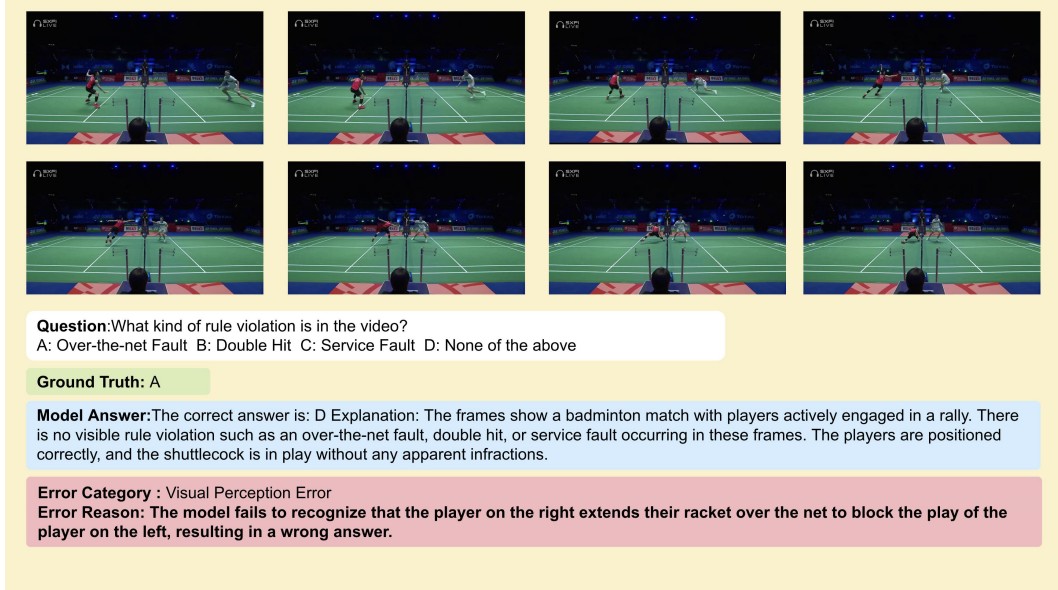

**Question:** What kind of rule violation is in the video?
A: Over-the-net Fault  B: Double Hit  C: Service Fault  D: None of the above

**Ground Truth:** A

**Model Answer:** The correct answer is: D Explanation: The frames show a badminton match with players actively engaged in a rally. There is no visible rule violation such as an over-the-net fault, double hit, or service fault occurring in these frames. The players are positioned correctly, and the shuttlecock is in play without any apparent infractions.

**Error Category :** Visual Perception Error
**Error Reason: The model fails to recognize that the player on the right extends their racket over the net to block the play of the player on the left, resulting in a wrong answer.**

Figure 36: Examples of Visual Perception Error and Error Reason Explanation

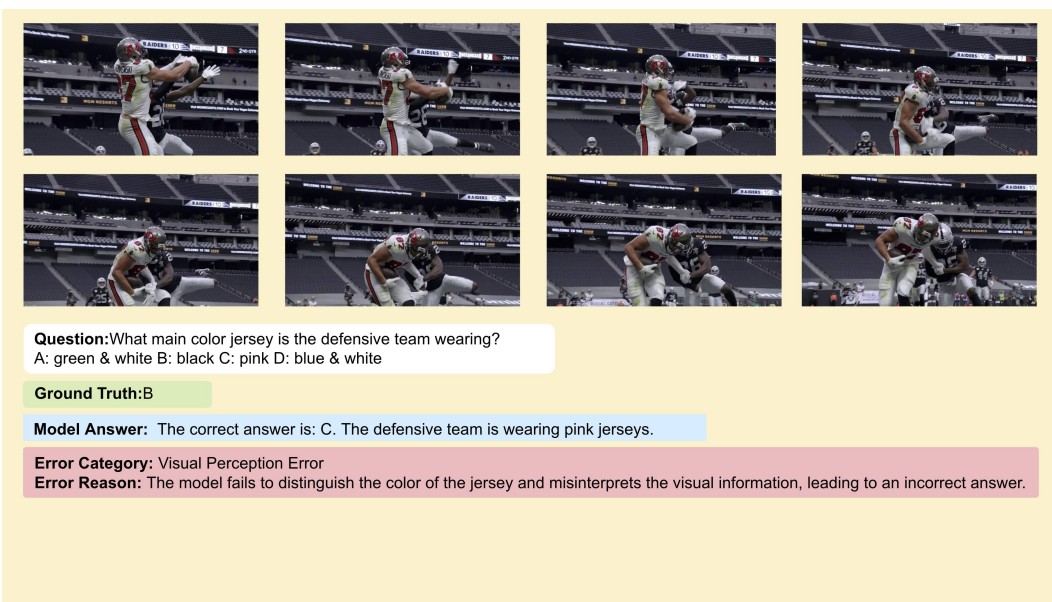

Figure 37: Examples of Visual Perception Error and Error Reason Explanation

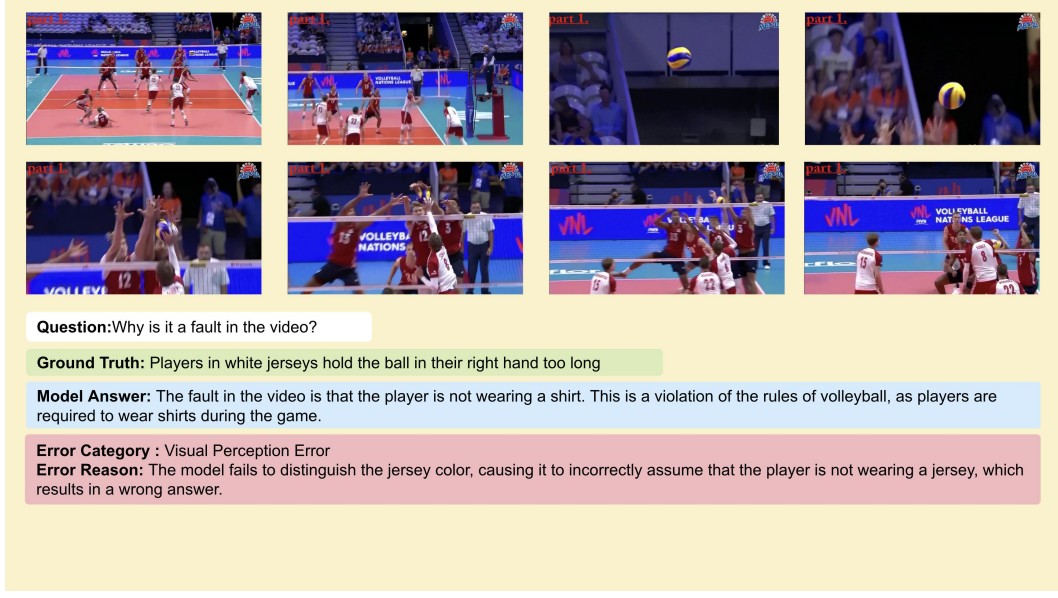

Figure 38: Examples of Visual Perception Error and Error Reason Explanation

## Q.3 HALLUCINATION ERROR

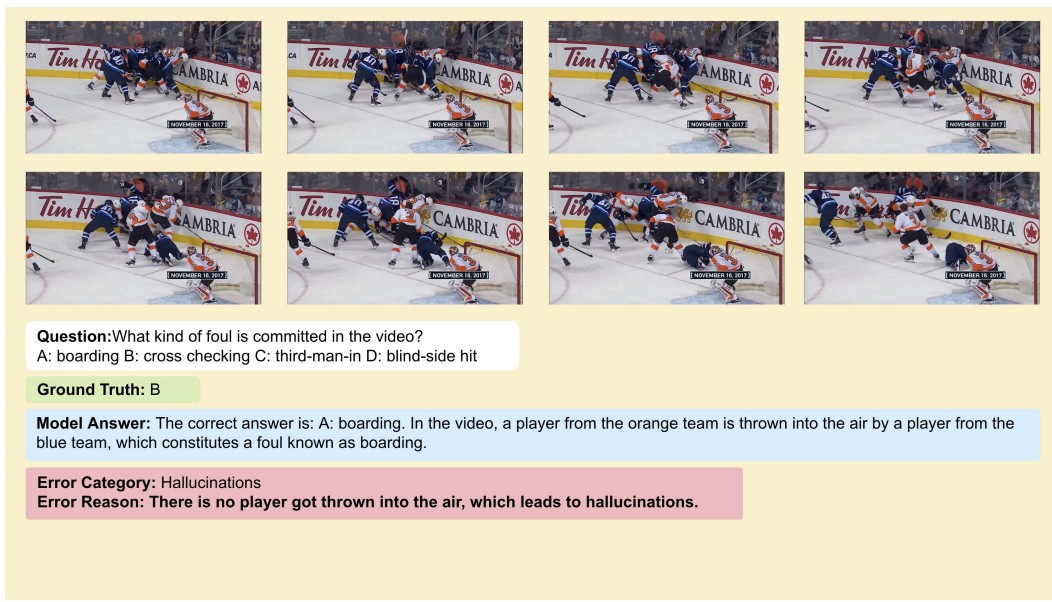

Figure 39: Examples of Hallucinations Error and Error Reason Explanation

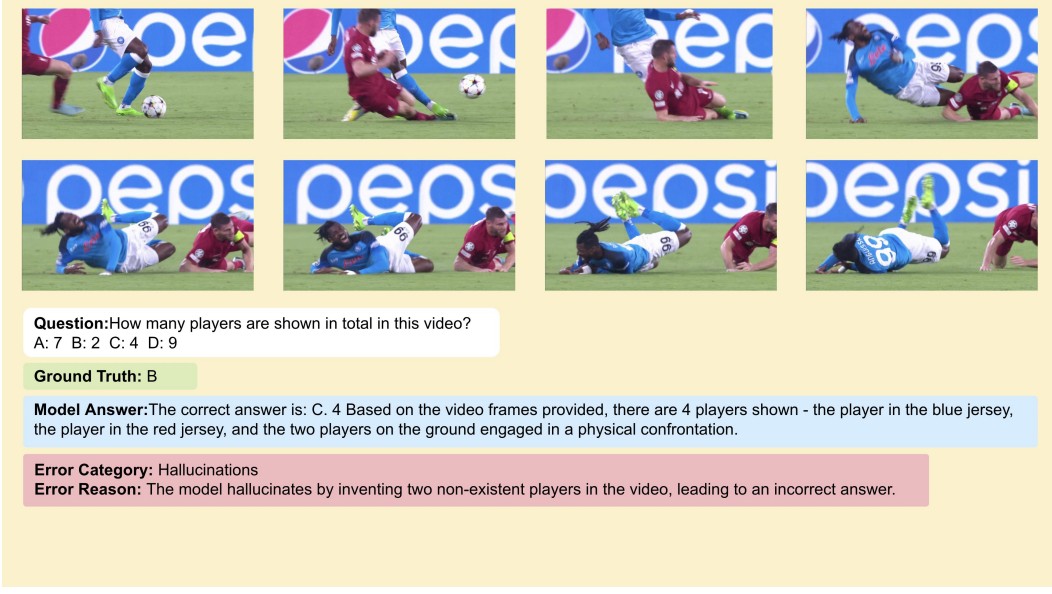

Figure 40: Examples of Hallucination Error and Error Reason Explanation

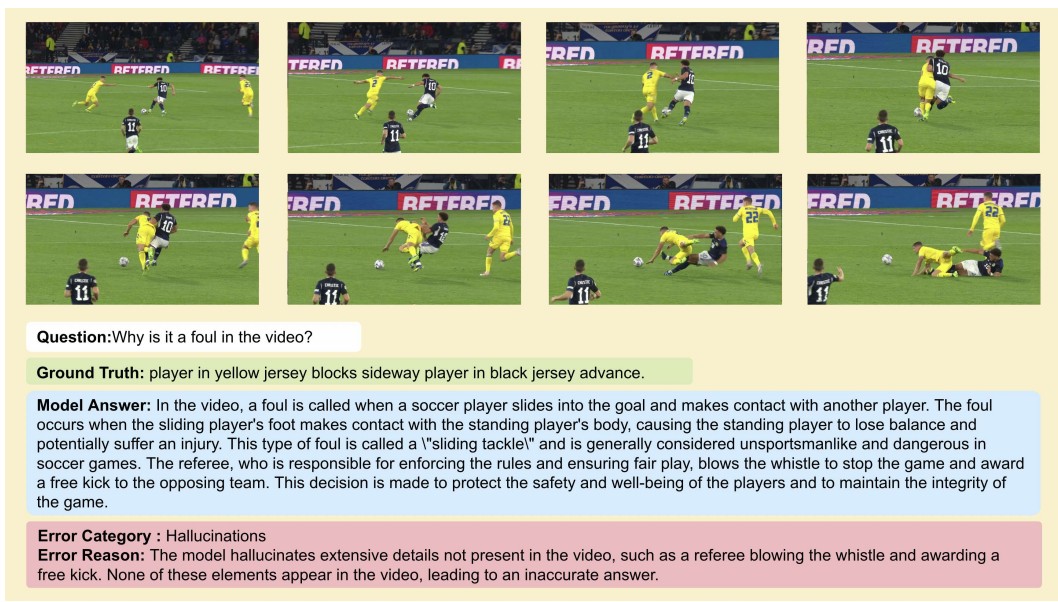

Figure 41: Examples of Hallucination Error and Error Reason Explanation

## Q.4    REASONING ERROR

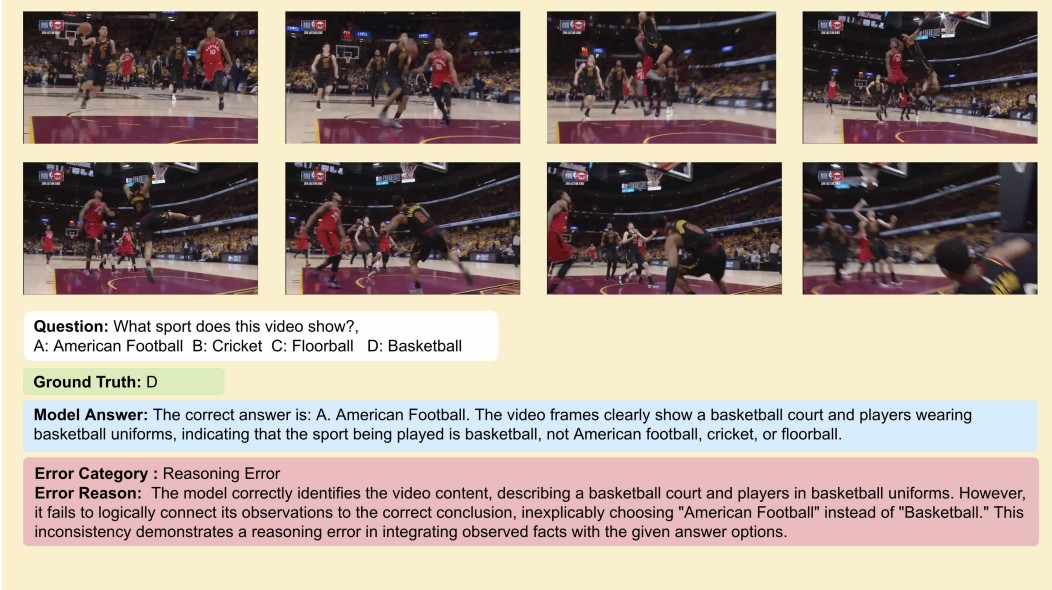

Figure 42: Examples of Reasoning Error and Error Reason Explanation

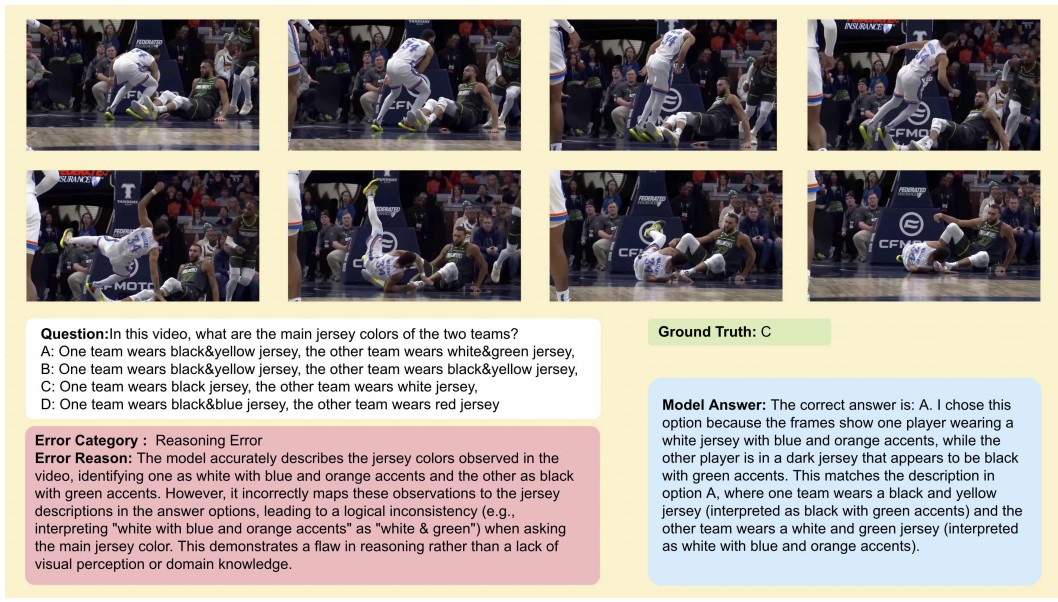

Figure 43: Examples of Reasoning Error and Error Reason Explanation

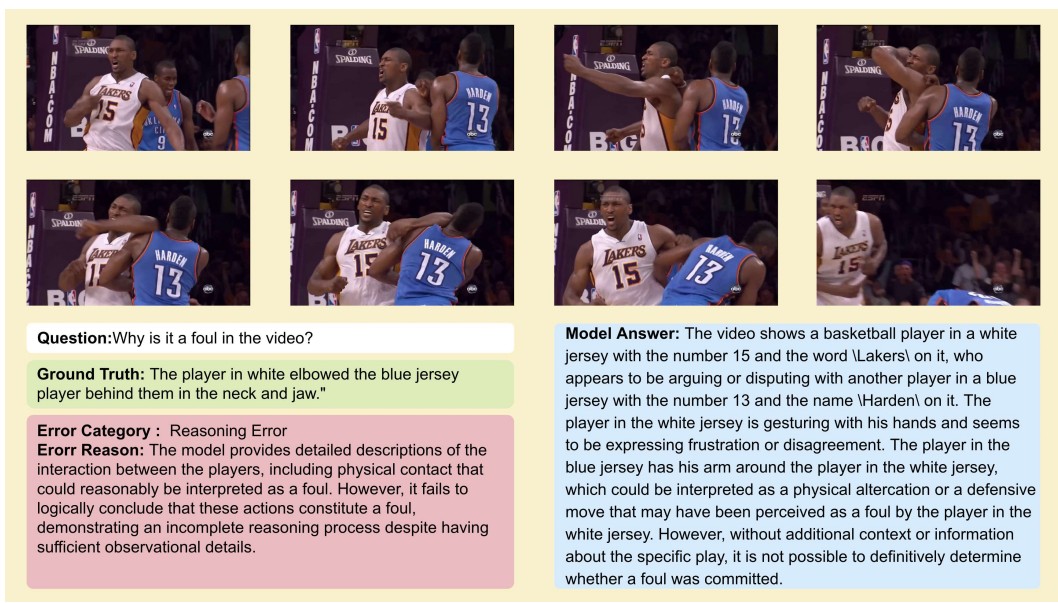

Figure 44: Examples of Reasoning Error and Error Reason Explanation

Q.5 LACK OF DOMAIN KNOWLEDGE

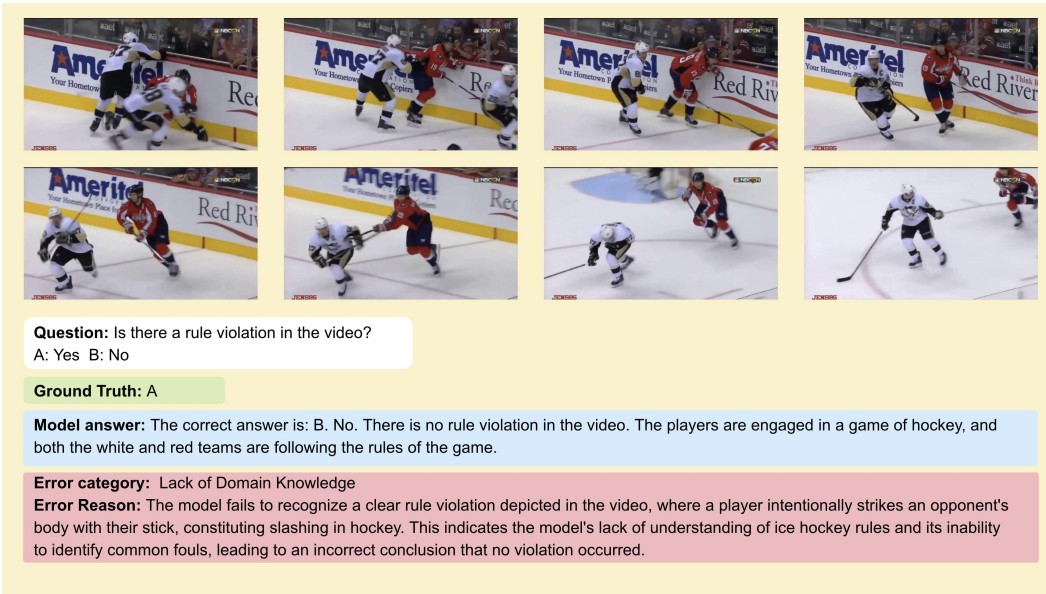

Figure 45: Examples of Lack of Domain Knowledge and Error Reason Explanation

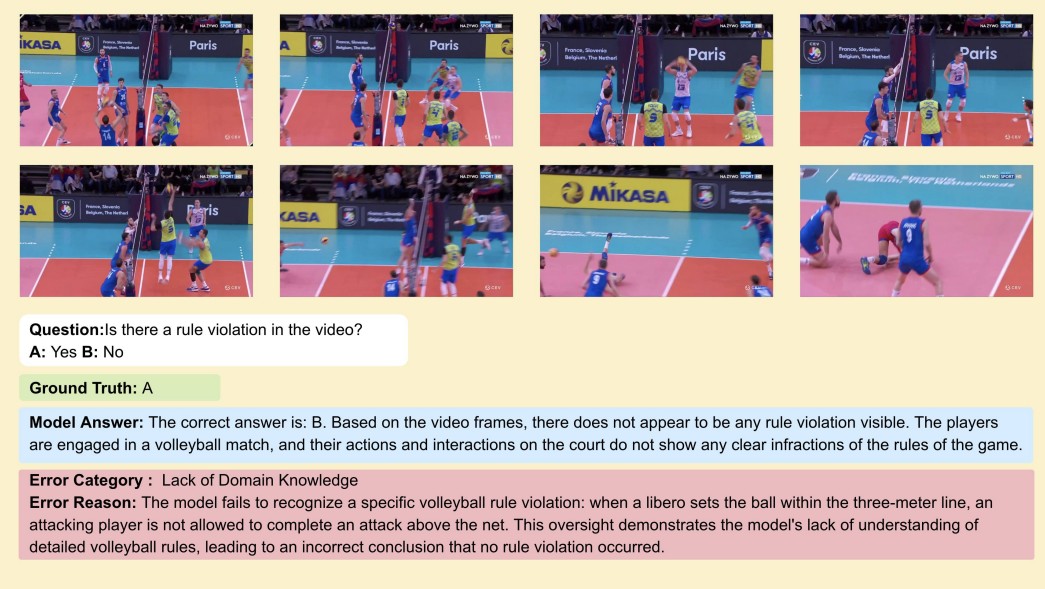

Figure 46: Examples of Lack of Domain Knowledge and Error Reason Explanation

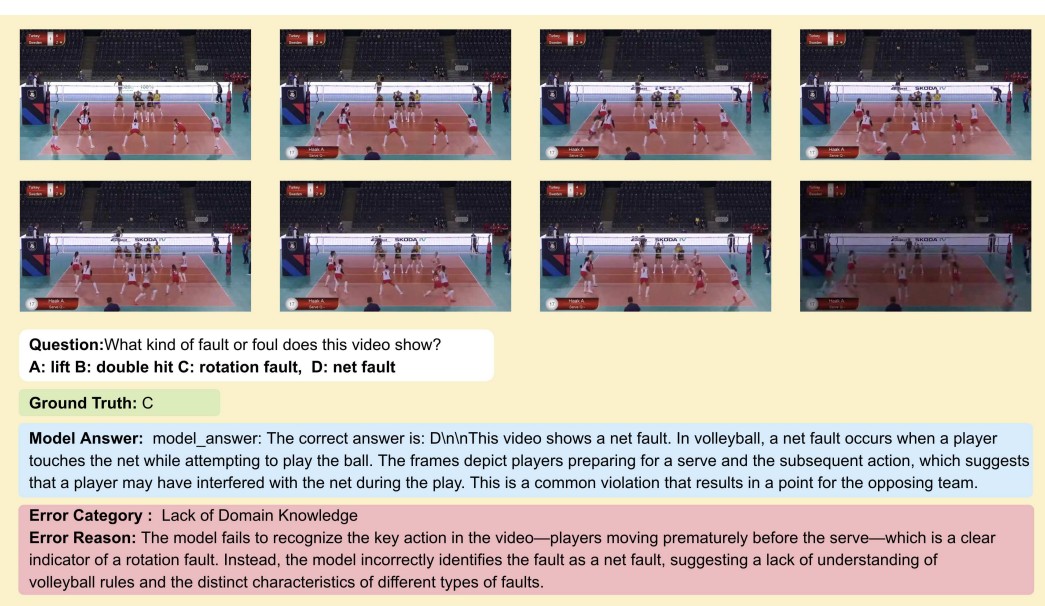

Figure 47: Examples of Lack of Domain Knowledge and Error Reason Explanation

