# OpenReview forum: "SPORTU: A Comprehensive Sports Understanding Benchmark for Multimodal Large Language Models"
_ICLR.cc/2025/Conference — ICLR 2025 Poster_

### Official Review · Reviewer_rN5c · 2024-11-03

**Soundness:** 2
**Presentation:** 2
**Contribution:** 2
**Rating:** 5
**Confidence:** 4

**Summary:**

This paper prensents SPORTU, a comprehensive Sports Understanding Benchmark that integrates both text-based and video-based tasks to evaluate models’ sports reasoning and knowledge application capabilities. Based on this benchmark, this paper tests the capability of existing open-source or close-source  models.

**Strengths:**

1. As a sport domain understanding benchmark, the proposed  SPORTU combines text-based and video-based tasks to assess models' sports reasoning and knowledge application abilities.

2. The evaluation setting is comprehensive including the direct prompting, chain-of-thought (CoT) prompting. In addition, few-shot promoting is also applied in SPORT-text evaluation.

**Weaknesses:**

1. Unclear motivation. The authors should clarify the differences between the proposed SPORTU and existing sport domain understanding benchmarks. Although discussions have been made in introduction and related work section together with Table 1, it is still unclear why the introduced features , for example, slow motion, multi-camera angles are important. More discussions and visualizations are needed.

2. Missing details in dataset construction. There exist some unclear details in the dataset construction. For example, how to guarantee the multi-camera setting? Is it achieved simply by human annotator check? In addiation, the proposed SPORTU contains both the multi-choice and open-ended question, how are these two categories divided?

3. More advanced evaluation methods should be applied. For example, ST-LLM [1], qwen-vl [2]

4. The paper writing should be polished. Some references are missing, for example "Section ??" in Line 821. The quotation mark error in '”Why is it a foul in the video?”' in Linee 482.

[1] ST-LLM: Large Language Models Are Effective Temporal Learners
[2] Qwen-VL: A Versatile Vision-Language Model for Understanding, Localization, Text Reading, and Beyond

**Questions:**

Please refer to the weakness part.

---

> ### Author Response · Authors · 2024-11-24
> **Response to Reviewer rN5c (Part1/2)**
>
> Dear Reviewer rN5c,
>
> We sincerely appreciate your valuable feedback and suggestions, which have significantly contributed to improving our work and broadening our evaluation. Below, we address each of your comments and questions:
>
> **About the motivation and importance of SPORTU:**
>
> First, we want to reaffirm the motivation behind SPORTU. Sports rules are a critical and indispensable part of sports, designed to protect players, ensure fairness, and enhance the viewing experience. These rules vary significantly across different sports, making them a key aspect of sports understanding. As MLLMs begin to demonstrate reasoning capabilities in various domains, sports remain an under-explored area. MLLMs with advanced reasoning capabilities can help non-experts quickly grasp sports rules and dynamics, enabling them to better enjoy and understand games.
> Existing works, as discussed in our introduction and related work, either lack depth by not incorporating detailed rule-related questions—an essential component for evaluating sports understanding—or focus on a single sport, limiting their scope. Evaluating MLLMs' capabilities in just one sport is insufficient to demonstrate comprehensive sports understanding across diverse disciplines. This is why we proposed SPORTU, a benchmark designed to address these gaps by covering multiple sports and incorporating tasks that require detailed reasoning. To enhance the dataset's diversity and quality, we introduced slow-motion and multi-camera angle videos. These features are essential for evaluating MLLMs in practical and realistic conditions:
>
> * **Slow Motion**: In real-world sports broadcasts, fouls and rule violations are often difficult to identify in real-time due to normal-speed, wide-angle shots. Critical details, such as subtle hand or foot movements, are typically captured through slow-motion replays. By including slow-motion clips, we ensure that models are tested on fine-grained action recognition, which is critical for sports reasoning tasks.
> * **Multi-Camera Angles**: In professional sports, rule violations are often reviewed from multiple angles during video replays to ensure accurate judgment. Multi-camera angles in our dataset enable us to test whether models can maintain consistency in reasoning and predictions across different perspectives. This is especially valuable for evaluating MLLMs’ robustness and understanding in dynamic scenarios.
>
> We recognize the need to provide more discussions and visualizations to emphasize these motivations. To address this, we have added examples in Appendix D to demonstrate cases where the same foul yields different model outputs depending on the angle.
>
> Additionally, Section 3 have been updated to explicitly discuss the rationale for including slow-motion and multi-camera videos. We also added a Broader Impacts section in Appendix B to discuss the border impacts of our benchmark.
>
>
> **About missing details in dataset construction:**
>
> Our expert annotators manually cropped video clips from replay footage to include multiple perspectives of the same foul. Since such replays are standard in sports broadcasts, this process ensures accuracy while minimizing additional manual work.
> As for the division between multiple-choice and open-ended questions, this is explained in Sections 3.2 and 3.3, but we have revised these sections to make the process clearer.
> * **Multiple-Choice Questions**: Annotators labeled the category of the ground truth such as foul (e.g., "handball" or "offside") based on the video content. These labels were then used to generate multiple-choice questions with the ground truth as the correct answer and distractors derived from other categories.
> * **Open-Ended Questions**: Annotators provided detailed explanations for the rule violation or foul observed in the video. These explanations, instead of simple labels, were used as the ground truth for generating open-ended questions, ensuring the model is tested on reasoning and explanation generation.

---

> ### Author Response · Authors · 2024-11-24
> **Response to Reviewer rN5c (Part2/2)**
>
> **About more advanced evaluation methods:**
>
> We greatly appreciate your suggestion to include ST-LLM and Qwen-VL in our evaluations, which helped us gain more insights. We have expanded our experiments to include these two models. We listed these two models' performance below. You can also refer to our Result Section and Appendix G.  Additionally, the results are listed in the response of 7Lbr.
> | **Model** | **X-YR** | **X-RY** | **X-Y** |
> |-----------|----------|-----------|----------|
> | ST-LLM | 41.59 | 40.09 | **46.39** |
> | Qwen2-VL-72B | 69.18 | 62.65 | **70.94** |
>
> | Model | ROUGE-L | BERTScore | BLEURT | CTC Preservation | GPT-based G-Eval | Gemini-based G-Eval | Claude-based G-Eval | Average G-Eval | Human Rating* |
> |-------|----------|------------|---------|------------------|------------------|-------------------|-------------------|----------------|---------------|
> | Qwen2-VL-72B | 0.10 | 0.42 | 0.39 | 0.41 | 1.62 | 1.94 | 1.72 | 1.76 | 2.08 |
> | ST-LLM | 0.13 | 0.38 | 0.20 | 0.41 | 1.30 | 1.50 | 1.52 | 1.44 | 1.22 |
>
> Along with feedback from other reviewers, we explored two more prompt settings to analyze the reasoning capabilities of the models further. These supplementary experiments provide an even more comprehensive evaluation and insights into our work. We hope these updates address your concerns and enhance our overall contribution to the field.
>
> **About paper writing:**
>
> Thank you for your detailed review. We have corrected typos and carefully reviewed the entire paper to address similar issues.
>
> We are grateful for your thoughtful feedback again. Please let us know if there are any additional concerns or points you would like us to address.

---

> ### Author Response · Authors · 2024-11-25
> **Request to Reviewer rN5c**
>
> Dear Reviewer rN5c,
>
> We kindly remind you that the rebuttal period is about to close, and we have made our best efforts to address your concerns. If they appear satisfactory, we kindly request that you take a moment to revisit your scores and share your updated thoughts. Please feel free to share if you have any follow-up questions, and we will make our best effort to handle them before the rebuttal period concludes.
>
> Thank you,
>
> Authors

---

> ### Author Response · Authors · 2024-11-28
> **Response to Reviewer rN5c**
>
> Dear Reviewer rN5c,
>
> Thank you for your reply. We would like to clarify that our benchmark has distinctive features that are absent in previous datasets, making significant contributions to evaluating and understanding MLLMs' deep sports understanding capabilities.
>
> - **Integration of rules and actions:** SPORTU uniquely ties video-based tasks to sports rules, requiring models to reason about how actions in different contexts lead to distinct rule applications or violations, rather than just extending from single-motion scenarios to multi-motion ones. In addition, understanding rules is the most important and foundational part of sports.
>
>
> - **Beyond surface-level questions:** Unlike existing benchmarks, SPORTU goes beyond identifying motions or sports and challenges models to answer complex questions such as "Why is this a foul?"—requiring both fine-grained action recognition and rule-based reasoning.
>
>
> - **High-quality human annotations:** We have especially added high-quality human annotated explanations in the SPORTU-text part and SPORTU-video open-ended part. This ensures high-quality video ground truth and question answering, which are critical for evaluating fine-grained tasks like foul detection.
>
>
> - **Comprehensive evaluation of sports understanding:** SPORTU is the first benchmark explicitly designed to evaluate MLLMs’ sports understanding across multiple sports and aspects, through multi-level difficulty questions, and pushes evaluation in the sports domain further.
>
> As sports continue to captivate people worldwide, MLLMs have the potential to help users engage more deeply with this domain. A benchmark to test MLLMs' understanding of sports is therefore essential. The evaluation of SPORTU clearly highlights the current limitations of MLLMs in deep sports reasoning, emphasizing the need for further advancements in this area.
>
> We hope this clarifies our contributions and look forward to addressing any further questions or concerns.

---

> ### Author Response · Authors · 2024-12-01
> **Follow-Up Request for Reviewer rN5c**
>
> Dear Reviewer rN5c,
>
> Thank you for your previous response again. As the rebuttal period is approaching its end, we would like to kindly follow up on the clarifications regarding our contributions that we provided in our earlier response. We hope these clarifications effectively highlight the distinctions between existing sports benchmarks and ours.
>
> Please let us know if you have further questions or require clarifications. We are more than happy to address them promptly within the remaining time.
>
> Thank you once again for your valuable insights and for helping us improve our paper.
>
> The Authors

---

### Official Review · Reviewer_gCvg · 2024-11-04

**Soundness:** 4
**Presentation:** 4
**Contribution:** 4
**Rating:** 6
**Confidence:** 4

**Summary:**

In the AI+Sports area, existing works are limited to restricted kinds of sports, absence of explanations, or lack of reasoning on rules, and it proposes SPORTU consisting of SPORTU-text and SPORTU-video to boost understanding more sports with rules understanding. SPORTU-text evaluates models on rule comprehension and strategy understanding in the pure text domain and SPORTU-video evaluates models on understanding both video details and rules in the video domain.
It evaluates LLMs and MLLMs on SPORTU-text and SPORTU-video, revealing their limitations in complicated sports questions.

**Strengths:**

1. It proposes SPORTU-text and SPORTU-video to boost understanding more sports with rules understanding in text and video domains.
2. It analyzes the views, reasoning prompts, sport types, the error types, which are comprehensive.
3. The writing is clear.

**Weaknesses:**

1. A benchmark aims to evaluate certain abilities and give some insights. The paper does not deeply discuss why the models have different performances and does not give advice on how to resolve the problem of understanding videos and reasoning on rules.

2. Prompt strategy in LLM can also be tested on MLLM when evaluating on SPORTU video benchmark to see how the reasoning process influences MLLM.

3. It's not very clear if the questions in this dataset can comprehensively detect the models' abilities to understand sports.

4. The Pearson correlation between humans and the other metrics is low. Many are near 0.

**Questions:**

1. How do you split the SPORTU-text questions into rules-related, strategy-related, and scenario-related? What is your basis?

2. What are the results on rule/strategy/scenario, respectively, on sportu-text?

3. How is the error analysis in 5.1 conducted? Is there a definition for each error type?

---

> ### Author Response · Authors · 2024-11-24
> **Response to Reviewer gCvg (Part 1)**
>
> Dear Reviewer  gCvg:
>
> We sincerely thank you for your detailed and constructive feedback. Your insights have helped us conduct more comprehensive experiments and bring valuable new perspectives to the paper. Below, we address your concerns and questions in detail.
>
> **About the discussion of varying model performance and advice on resolution:**
>
> In the Result Section, we have mentioned models that tend to perform well on easy-level questions but show a significant gap in hard-level questions, indicating a lack of domain knowledge, particularly in rule comprehension.
> We also found that most proprietary models, along with the Qwen2-VL-72B model, achieved higher G-Eval and human rating scores than the 7B open-source models. This indicates that proprietary models exhibit stronger reasoning abilities than the 7B open-source models. However, none of the MLLMs achieved an average G-Eval score above 3, highlighting a gap in the deep domain knowledge required for reasoning in video-based sports understanding.  We have added this finding in the result section.
> Regarding the resolution, We have added a Discussion section in Appendix C. We think better frame extraction strategies need to be developed specifically for sports tasks to ensure that the actions critical for reasoning about the question are clearly provided to the model. A more effective grounding method could also enhance the model’s ability to distinguish actions more accurately. During the experiments, we noticed that while models sometimes captured and described most of the correct movements, they often failed to infer the corresponding rule violations. This highlights a significant gap in connecting recognized actions to specific rules, underscoring the need for models to not only identify movements but also to understand what a rule violation should look like.
>
> **About testing prompt strategy in LLM on MLLM:**
>
> We appreciate this suggestion. We have tested additional prompt strategies on MLLMs for SPORTU-video. Specifically, we added two prompt strategies, including X → Y , X → RY. The results show that X → Y achieved the highest overall performance in most of the models, outperforming X → Y R and X → RY. For most models, the accuracy follows the order: X → Y > X → Y R > X → RY. However, as task difficulty increases, especially in hard-level questions that require more reasoning, performance drops significantly for X → RY due to hallucinated rationales, which mislead the model’s answer inference.
> For example, Qwen72b achieved 70.95% accuracy in the X → Y setting but dropped to 69.17% in X → Y R and further to 62.51% in X → RY. On hard-level questions, the drop was particularly significant, from 44.12% (X → Y) to 30.55% (X → RY). This highlights the challenges of reasoning-based prompts in complex tasks. This finding has also been introduced in Zhang's work[1]. We have incorporated these insights into Appendix G and H, including additional examples that demonstrate cases where X → Y succeeded while X → RY failed due to hallucinated rationales.  The detailed results are presented in the Results section. Additionally, the results are listed in the response of 7Lbr.
>
> **About whether the questions in this dataset can comprehensively detect the models' abilities to understand sports:**
>
> Our dataset is the first benchmark that comprehensively evaluates models' fundamental sports understanding across three distinct difficulty levels. Easy-level questions are designed for individuals with general commonsense and no specific sports knowledge, medium-level questions require foundational sports knowledge beyond commonsense, and hard-level questions target those with an in-depth understanding of specific sports, including sport-specific rule comprehension. This tiered structure provides a comprehensive evaluation of MLLMs' sports comprehension across varying levels of complexity.
>
> **About the low correlation between human and automatic metrics:**
>
> The low Pearson correlation between human evaluations and other metrics highlights the limitations of current evaluation methods for sports-related reasoning tasks. Unlike general domains, sports explanations often rely on precise terminology and detailed action descriptions, where even minor differences can lead to entirely different outcomes. This demonstrates the need for sports-specific evaluation metrics that better capture the nuances of rule-based reasoning. We believe this is an important future direction and a critical step toward developing automatic metrics that align with human judgments.
>
> [1] Z. Zhang et al., “Multimodal chain-of-thought reasoning in language models” TMLR

---

> ### Author Response · Authors · 2024-11-24
> **Response to Reviewer gCvg (Part 2)**
>
> **About your questions:**
>
> * _How do you split the SPORTU-text questions?_
>
> Our expert team manually annotated and labeled the questions into three categories: 1) rule-related questions focus on explicit sports rules; 2) strategy-related questions involve tactical or strategic decisions; 3) scenario-related questions provide specific contexts or player interactions (e.g., "Player A performs action X, and Player B reacts"). While some questions may overlap categories, our experts assigned each question to the most appropriate category based on their domain knowledge.
>
> * _What are the results on rule/strategy/scenario, respectively, on sportu-text?_
> The classification results are provided below and have also been included in Appendix E and Appendix F for further reference.
>
> | **Model** | **Acc.(%)** | **ROUGE-L** | **BERTScore** | **BLEURT** | **CTC Presv.** | **G-Eval** |
> |-----------|-------------|-------------|---------------|------------|----------------|-------------|
> | **Overall** |||||||
> | Claude-3.5-Sonnet | 64.67 | 0.26 | 0.65 | 0.57 | 0.43 | 3.78 |
> | Gemini-1.5 | 62.67 | 0.28 | 0.62 | 0.53 | 0.43 | 3.79 |
> | GPT-4o | 68.78 | 0.27 | 0.66 | 0.57 | 0.43 | 4.16 |
> | Llama3.1-405B | 64.44 | 0.25 | 0.64 | 0.55 | 0.43 | 3.89 |
> | **Rule** |||||||
> | Claude-3.5-Sonnet | 64.67 | 0.26 | 0.65 | 0.57 | 0.43 | 3.79 |
> | Gemini-1.5 | 62.67 | 0.28 | 0.62 | 0.53 | 0.42 | 3.80 |
> | GPT-4o | 68.78 | 0.27 | 0.66 | 0.57 | 0.43 | 4.16 |
> | Llama3.1-405B | 64.44 | 0.25 | 0.64 | 0.55 | 0.43 | 3.89 |
> | **Strategy** |||||||
> | Claude-3.5-Sonnet | 67.43 | 0.25 | 0.65 | 0.57 | 0.43 | 3.89 |
> | Gemini-1.5 | 62.29 | 0.26 | 0.61 | 0.52 | 0.42 | 3.98 |
> | GPT-4o | 73.14 | 0.27 | 0.66 | 0.56 | 0.43 | 4.23 |
> | Llama3.1-405B | 65.71 | 0.24 | 0.63 | 0.54 | 0.43 | 3.98 |
> | **Scenario** |||||||
> | Claude-3.5-Sonnet | 66.20 | 0.26 | 0.65 | 0.57 | 0.43 | 3.86 |
> | Gemini-1.5 | 61.13 | 0.27 | 0.62 | 0.53 | 0.43 | 3.71 |
> | GPT-4o | 68.73 | 0.27 | 0.66 | 0.56 | 0.43 | 4.10 |
> | Llama3.1-405B | 66.57 | 0.25 | 0.64 | 0.55 | 0.43 | 3.91 |
>
> * _How is the error analysis in 5.1 conducted? Is there a definition for each error type?_
>
> As described in Section 5.1, our expert team selected 20 incorrect examples per sport for both multiple-choice and open-ended questions (20 for each) for each model. Since we have now included two additional models, bringing the total to 14, the total error cases have been updated to 3920, and the paper has been revised accordingly. The detailed definitions for each error type are included in Appendix M , along with representative examples.
>
> We are grateful for your thoughtful feedback, which has helped us improve both the clarity and depth of our work. Please let us know if there are any additional concerns or points you would like us to address.

---

> ### Author Response · Authors · 2024-11-25
> **Request to Reviewer gCvg**
>
> Dear Reviewer gCvg,
>
> We kindly remind you that the rebuttal period is about to close, and we have made our best efforts to address your concerns. If they appear satisfactory, we kindly request that you take a moment to revisit your scores and share your updated thoughts. Please feel free to share if you have any follow-up questions, and we will make our best effort to handle them before the rebuttal period concludes.
>
> Thank you,
>
> Authors

---

> ### Author Response · Authors · 2024-12-01
> **Follow-Up Request for Reviewer gCvg**
>
> Dear Reviewer gCvg,
>
> As the rebuttal period is approaching its end, we would like to kindly follow up regarding our responses to your comments. We have tried our best to carefully address your concerns and would greatly appreciate it if you could share any additional feedback or consider adjusting your score.
>
> Please let us know if you have further questions or require clarifications. We are more than happy to address them promptly within the remaining time.
>
> Thank you once again for your valuable insights and for helping us improve our paper.
>
> The Authors

---

> ### Comment · Reviewer_gCvg · 2024-12-03
> **Reply to Authors**
>
> Thank all the authors for addressing my concerns in this rebuttal and it addresses my concerns. After taking other reviews and rebuttals into consideration, I maintain my score.

---

### Official Review · Reviewer_uNov · 2024-11-04

**Soundness:** 2
**Presentation:** 2
**Contribution:** 2
**Rating:** 6
**Confidence:** 4

**Summary:**

The paper provides a multimodal dataset (text and slow-motion video) for evaluating (multimodal) LLM capabilities in the sports domain.

**Strengths:**

- A multimodal new dataset for sports domain (with multiple sports) and well annotated by experts; the dataset should be helpful for the research communities

- A well prompting capabilities to show the limitation of current LLM capabilities on the dataset.

- Evaluating several reasonable public or private LLM models.

**Weaknesses:**

- Yet another vertical dataset for LLM
- It's helpful but marginal to expand the technical depth for the community
- not clearly identified what current models failed.

**Questions:**

- The video quality for the datasets
- For each sports type, the video are biased to certain views or events?

---

> ### Author Response · Authors · 2024-11-24
> **Response to Reviewer uNov**
>
> Dear Reviewer uNov,
>
> Thank you for your valuable feedback. We appreciate your insights and have addressed your concerns as follows:
>
> **Yet another vertical dataset for LLM:**
> Our dataset offers unique challenges and insights, making it a critical benchmark for advancing MLLMs in sports. Specifically, sports understanding requires models to integrate fine-grained action recognition with complex rule-based reasoning, a challenge that extends beyond typical domain-specific tasks. SPORTU introduces the first-ever comprehensive benchmark combining text and slow-motion video tasks across various sports. It is designed to evaluate MLLMs' ability to connect visual perception with sports-specific rule comprehension.
>
> **About the technical depth:**
> Our primary contribution is to introduce two new sports datasets that are more challenging than existing datasets and contain explanations. We benchmark LLMs on these datasets, demonstrating a significant gap in their performance on hard-level questions that require understanding movement and matching it with corresponding rules. The goal is to facilitate future research on integrating MLLMs for sports. While technical depth is present, it is not the main focus of this paper.
>
> **To clarify what current models failed:**
> We conducted additional analyses, including examples of hallucinated rationales and reasoning errors in complex foul scenarios. We also added the results for X-RY and X-Y in the Results section. Our results show that models struggle with reasoning-heavy tasks, where precise action recognition must be linked with domain-specific rules. These examples are included in Appendix H, and a discussion section in Appendix C is added to demonstrate why robust rationale generation is critical in the sports domain. We believe that the insights from these failures are essential for guiding future improvements in MLLM research.
>
> **About your questions:**
>
> * The video quality for the datasets:
> The quality of the videos in our dataset has been rigorously controlled through three key aspects:
>   * Annotator Expertise: Our annotation team consisted of nine experts, including two intercollegiate student-athletes with over 12 years of experience and seven players with at least five years of training in their respective sports. This ensured deep domain knowledge for accurate annotations.
>   * Video Cropping: All videos were manually cropped by our expert annotators to ensure they focus on key actions relevant to the designed questions. This process guarantees that each video provides sufficient clarity and context for detailed analysis.
>   * Verification Protocol: Annotators double-reviewed all collected videos to ensure accuracy and quality. We also prioritized the removal of controversial or hard-to-interpret clips to minimize potential mislabeling.
> These measures collectively ensure that the dataset meets high-quality standards, suitable for detailed action analysis and rule comprehension tasks. And This process is shown in section 3.1.
>
>
> * For each sports type, the videos are biased to certain views or events?
>      * We do not believe the videos are biased toward certain views or events. The dataset includes videos featuring common rule violations that occur across all levels of games and competitions. Furthermore, all videos were sourced from a variety of different competitions to ensure diversity in scenarios and perspectives.
>
> We are grateful for your thoughtful feedback, which has helped us improve both the clarity and depth of our work. Please let us know if there are any additional concerns or points you would like us to address.

---

> ### Author Response · Authors · 2024-12-01
> **Follow-Up Request for Reviewer uNov**
>
> Dear Reviewer uNov,
>
> As the rebuttal period is approaching its end, we would like to kindly follow up regarding our responses to your comments. We have tried our best to carefully address your concerns and would greatly appreciate it if you could share any additional feedback or consider adjusting your score.
>
> Please let us know if you have further questions or require clarifications. We are more than happy to address them promptly within the remaining time.
>
> Thank you once again for your valuable insights and for helping us improve our paper.
>
> The Authors

---

### Official Review · Reviewer_7Lbr · 2024-11-11

**Soundness:** 1
**Presentation:** 2
**Contribution:** 2
**Rating:** 5
**Confidence:** 4

**Summary:**

This paper introduces SPORTU, a new benchmark designed to evaluate the capabilities of Multimodal Large Language Models (MLLMs) in sports understanding and reasoning. SPORTU consists of two components: SPORTU-text, focusing on text-based reasoning, and SPORTU-video, focusing on video-based reasoning. The authors evaluate various LLMs on both components, revealing limitations in complex reasoning tasks.

**Strengths:**

1. The proposed dataset could be useful for the community.
2. Both close and open-sourced models are evaluated.
3. Metrics are studied with human verification.

**Weaknesses:**

1. The reviewer is concerned about the lack of diversity and coverage of the dataset because of the limited prompt templates and number of samples.

2. Implementation could be possibly flawed.
- The error in Figure 6 looks suspicious and makes the reviewer wonder whether the model is called correctly or not.
- The reasoning prompt asks the model to first generate answer and then reasoning, which is not optimal since the model's final answer cannot benefit from the reasoning process.
- It is known that LLM usually prefers its own answer so it is important to understand G-eval' quality with different LLMs as the rater.

Minor:
L821 typos of "Section ??"

**Questions:**

Please check weakness for details.

---

> ### Author Response · Authors · 2024-11-24
> **Response to Reviewer 7Lbr (Part 1/3)**
>
> Dear Reviewer 7Lbr,
>
> We really appreciate your constructive feedback, which helped us find more insights. Below, we address your concerns point by point.
>
> **About the diversity and coverage of the dataset:**
>
> Our dataset is the first to test MLLMs' sports understanding ability, particularly by combining action recognition with rule-based reasoning across different sports and varying difficulty levels. Existing datasets focus solely on action recognition or understanding the deep rules of a single sport. We argue that understanding actions alone does not equate to better sports understanding for MLLMs, as rules play a significant role in games and must also be comprehended. Moreover, performance on a single sport is insufficient to evaluate an MLLM's sports understanding across various sports, as rules differ significantly between sports. In contrast, our dataset is more diverse, covering multiple sports categories and difficulty levels. A comparison table with existing works is provided in the Related Work section. Notably, our dataset is the first to test MLLMs' ability to understand rules across various sports.
>
> We have added more templates and question examples in Appendix O to Q. Additionally, more questions and videos can be found in the anonymous link in the paper.
>
>
>
> **About implementations:**
> - _The error in Figure 6_: We appreciate your attention to the details of our error analysis. This error only appears a few times in the Gemini output, which we think is interesting to show an illustrative example of how templates can occasionally fail when other Gemini outputs work by using the same template. To provide a more comprehensive analysis, we have now added additional error examples in Appendix Q, covering a more comprehensive range of cases and models.
>
> - _The reasoning prompt_: We really appreciate your concern, which allows us to analyze this issue further.
> We have conducted supplementary experiments with two additional prompting strategies:
> X → Y: No-CoT, which directly predicts the answer.
> X → RY : Reasoning where answer inference is conditioned to the rationale.
> The results show that X → Y achieved the highest overall performance in most of the models, outperforming X → Y R and X → RY. For most models, the accuracy follows the order: X → Y > X → Y R > X → RY. However, as task difficulty increases, especially in hard-level questions that require more reasoning, performance drops significantly for X → RY due to hallucinated rationales, which mislead the model’s answer inference. For example, Qwen72b achieved 70.95% accuracy in the X → Y setting but dropped to 69.17% in X → Y R and further to 62.51% in X → RY. On hard-level questions, the drop was particularly significant, from 44.12% (X → Y) to 30.55% (X → RY). This highlights the challenges of reasoning-based prompts in complex tasks. This finding aligns with what is shown in Zhang’s work[1]. We have incorporated these insights into Appendix G and H, including additional examples that demonstrate cases where X → Y succeeded while X → RY failed due to hallucinated rationales. We attached the result below.
>
> [1] Z. Zhang et al., “Multimodal chain-of-thought reasoning in language models” TMLR

---

> ### Author Response · Authors · 2024-11-24
> **Response to Reviewer 7Lbr (Part 2/3)**
>
> **Three prompting strategies performance of MLLMs on SPORTU-video**
>
> The best results are **bolded**. The results highlight that models perform best with the X → Y prompt (25/56 leading performances), followed by X →YR (21/56), and X → RY(10/56).
>
> | Model | Difficulty | X-YR | X-RY | X-Y |
> |-------|------------|------|------|-----|
> | Claude-3.0-Haiku | Easy | **68.41** | 66.58 | 66.62 |
> | | Medium | 46.43 | 46.11 | **46.53** |
> | | Hard | 20.12 | 18.94 | **22.42** |
> | | Overall | **48.07** | 47.19 | 47.95 |
> | Claude-3.5-Sonnet | Easy | 88.65 | 63.83 | **89.15** |
> | | Medium | 65.10 | 55.52 | **65.88** |
> | | Hard | 52.57 | 39.32 | **53.06** |
> | | Overall | 69.52 | 55.08 | **70.18** |
> | Gemini 1.5 Pro | Easy | 85.85 | 85.20 | **87.52** |
> | | Medium | 58.25 | 58.11 | **61.22** |
> | | Hard | **43.53** | 42.75 | 39.98 |
> | | Overall | **65.13** | 63.04 | 64.93 |
> | Gemini 1.5 Flash | Easy | **85.99** | 59.07 | 85.19 |
> | | Medium | 53.38 | 50.85 | **58.56** |
> | | Hard | **38.73** | 12.89 | 38.26 |
> | | Overall | 59.97 | 46.68 | **62.52** |
> | GPT-4omini | Easy | 66.09 | 59.02 | **66.68** |
> | | Medium | 55.69 | 42.25 | **58.12** |
> | | Hard | **47.54** | 13.67 | 44.49 |
> | | Overall | 57.24 | 42.06 | **58.19** |
> | GPT-4o | Easy | **84.89** | 79.92 | 84.30 |
> | | Medium | 62.31 | **65.98** | 64.83 |
> | | Hard | **57.84** | 40.51 | 56.20 |
> | | Overall | 68.00 | 65.56 | **68.79** |
> | Qwen2-VL-72B | Easy | 94.86 | 84.16 | **95.11** |
> | | Medium | 66.27 | 61.69 | **66.97** |
> | | Hard | 36.53 | 30.56 | **44.12** |
> | | Overall | 69.18 | 62.65 | **70.94** |
> | ChatUniVi | Easy | **59.22** | 49.04 | 55.99 |
> | | Medium | **36.95** | 28.55 | 35.63 |
> | | Hard | 32.21 | 18.71 | **39.07** |
> | | Overall | **42.35** | 32.58 | 41.89 |
> | LLaVA-NeXT | Easy | **94.24** | 91.43 | 92.44 |
> | | Medium | **67.02** | 56.00 | 59.39 |
> | | Hard | 33.44 | **34.21** | 30.78 |
> | | Overall | **68.89** | 62.16 | 63.72 |
> | mPLUG-Owl3 | Easy | 87.28 | **88.51** | 87.11 |
> | | Medium | 55.58 | **58.75** | 57.37 |
> | | Hard | 25.40 | 24.88 | **28.89** |
> | | Overall | 59.26 | **61.27** | 60.80 |
> | ST-LLM | Easy | 59.26 | **68.54** | 63.09 |
> | | Medium | 36.08 | 33.23 | **41.14** |
> | | Hard | 30.51 | 22.10 | **36.08** |
> | | Overall | 41.59 | 40.09 | **46.39** |
> | Tarsier | Easy | **89.71** | 84.01 | 88.13 |
> | | Medium | **58.43** | 50.94 | 58.35 |
> | | Hard | 24.25 | 24.80 | **25.17** |
> | | Overall | **61.32** | 55.70 | 60.99 |
> | Video-ChatGPT | Easy | 62.36 | **63.91** | 37.04 |
> | | Medium | **39.40** | 38.64 | 36.01 |
> | | Hard | **32.55** | 19.12 | 22.82 |
> | | Overall | **44.63** | 42.36 | 34.05 |
> | VideoChat2 | Easy | 88.30 | 88.62 | **89.37** |
> | | Medium | 58.73 | **60.45** | 58.65 |
> | | Hard | 26.94 | **28.13** | 25.29 |
> | | Overall | 61.55 | **62.79** | 61.53 |

---

> ### Author Response · Authors · 2024-11-24
> **Response to Reviewer 7Lbr (Part 3/3)**
>
> -  _G-eval’s quality with different LLMs as the rater_: We appreciate your suggestion. To address this, we supplemented the G-eval evaluation with scores obtained using Claude 3.5 Sonnet, Gemini 1.5 Pro, and Llama 3.1-405B.   The result is shown below：
>
> **SPORTU-text Part**
>
> | Setting | Model | Accuracy(%) | ROUGE-L | BERTScore | BLEURT | CTC Preservation | GPT-based G-Eval | Gemini-based G-Eval | Claude-based G-Eval | Llama-based G-Eval | Average G-Eval |
> |---------|--------|-------------|----------|------------|---------|------------------|------------------|-------------------|-------------------|-------------------|----------------|
> | X→RY(0CoT) | Claude-3.5-Sonnet | 64.67 | 0.26 | 0.65 | 0.57 | 0.43 | 3.78 | 3.25 | 3.28 | 4.07 | 3.60 |
> | | gemini-1.5 Pro | 62.67 | 0.28 | 0.62 | 0.53 | 0.43 | 3.79 | 3.57 | 3.39 | 3.98 | 3.68 |
> | | GPT-4o | 68.78 | 0.27 | 0.66 | 0.57 | 0.43 | 4.16 | 3.42 | 3.43 | 4.37 | 3.85 |
> | | Llama3.1-405B | 64.44 | 0.25 | 0.64 | 0.55 | 0.43 | 3.89 | 3.19 | 2.74 | 3.90 | 3.39 |
> | X→RY(5CoT) | Claude-3.5-Sonnet | 65.22 | 0.27 | 0.65 | 0.56 | 0.43 | 3.98 | 3.43 | 3.39 | 4.15 | 3.74 |
> | | gemini-1.5 Pro | 61.22 | 0.30 | 0.62 | 0.53 | 0.43 | 3.73 | 3.51 | 3.49 | 3.38 | 3.53 |
> | | GPT-4o | 71.00 | 0.33 | 0.68 | 0.58 | 0.44 | 4.13 | 3.52 | 3.59 | 4.15 | 3.85 |
> | | Llama3.1-405B | 65.22 | 0.32 | 0.67 | 0.57 | 0.44 | 3.81 | 3.28 | 3.33 | 4.02 | 3.61 | CopyRetryClaude does not have the ability to run the code it generates yet.
>
> **SPORTU-video Part**
>
> | Model | ROUGE-L | BERTScore | BLEURT | CTC Preservation | GPT-based G-Eval | Gemini-based G-Eval | Claude-based G-Eval | Average G-Eval | Human Rating* |
> |-------|----------|------------|---------|------------------|------------------|-------------------|-------------------|----------------|---------------|
> | Claude-3.0-Haiku | 0.08 | 0.41 | 0.43 | 0.39 | 1.55 | 1.80 | 1.63 | 1.66 | 1.93 |
> | Claude-3.5-Sonnet | 0.05 | 0.40 | 0.43 | 0.39 | 1.62 | 1.89 | 1.59 | 1.70 | 2.13 |
> | Gemini 1.5 Pro | 0.08 | 0.38 | 0.36 | 0.38 | 1.11 | 1.16 | 1.20 | 1.16 | 1.19 |
> | Gemini 1.5 Flash | 0.13 | 0.45 | 0.42 | 0.39 | 1.34 | 1.70 | 1.62 | 1.55 | 1.84 |
> | GPT-4omini | 0.05 | 0.39 | 0.36 | 0.38 | 1.60 | 1.94 | 1.65 | 1.73 | 2.17 |
> | GPT-4o | 0.07 | 0.41 | 0.43 | 0.39 | 1.84 | 1.17 | 1.75 | 1.59 | 2.51 |
> | ChatUniVi | 0.07 | 0.39 | 0.37 | 0.38 | 1.27 | 1.39 | 1.45 | 1.37 | 1.48 |
> | LLaVA-NeXT | 0.17 | 0.47 | 0.38 | 0.40 | 1.47 | 1.63 | 1.75 | 1.61 | 1.62 |
> | mPLUG-Owl3 | 0.15 | 0.44 | 0.37 | 0.39 | 1.38 | 1.60 | 1.75 | 1.58 | 1.46 |
> | Tarsier | 0.12 | 0.45 | 0.36 | 0.40 | 1.36 | 0.70 | 1.78 | 1.28 | 1.63 |
> | Video-ChatGPT | 0.08 | 0.39 | 0.35 | 0.38 | 1.08 | 1.11 | 1.36 | 1.19 | 1.22 |
> | VideoChat2 | 0.23 | 0.49 | 0.35 | 0.40 | 1.43 | 1.73 | 1.79 | 1.65 | 1.48 |
> | ST-LLM | 0.13 | 0.38 | 0.20 | 0.41 | 1.30 | 1.50 | 1.52 | 1.44 | 1.22 |
> | Qwen2-VL-72B | 0.10 | 0.42 | 0.39 | 0.41 | 1.62 | 1.94 | 1.72 | 1.76 | 2.08 |
>
> **minor issues**, we corrected the typo and carefully reviewed the entire paper to ensure similar errors were addressed. Thank you for your careful review and for pointing them out.
>
> We are grateful for your thoughtful feedback, which has helped us improve both the clarity and depth of our work. Please let us know if there are any additional concerns or points you would like us to address.

---

> ### Author Response · Authors · 2024-11-25
> **Request to Reviewer 7Lbr**
>
> Dear Reviewer 7Lbr,
>
> We kindly remind you that the rebuttal period is about to close, and we have made our best efforts to address your concerns. If they appear satisfactory, we kindly request that you take a moment to revisit your scores and share your updated thoughts. Please feel free to share if you have any follow-up questions, and we will make our best effort to handle them before the rebuttal period concludes.
>
> Thank you,
>
> Authors

---

> ### Author Response · Authors · 2024-12-01
> **Follow-Up Request for Reviewer 7Lbr**
>
> Dear Reviewer 7Lbr,
>
> As the rebuttal period is approaching its end, we would like to kindly follow up regarding our responses to your comments. We have tried our best to carefully address your concerns and would greatly appreciate it if you could share any additional feedback or consider adjusting your score.
>
> Please let us know if you have further questions or require clarifications. We are more than happy to address them promptly within the remaining time.
>
> Thank you once again for your valuable insights and for helping us improve our paper.
>
> The Authors

---

### Author Response · Authors · 2024-11-24
**Official Comment by Authors**

Dear Reviewers,

We sincerely thank you for your valuable comments. We have updated the paper and highlighted the revisions according to your feedback in blue for your convenience. We also add more experiment results and examples in the appendix. We hope the updated version and this rebuttal address your concerns. Please feel free to share any additional feedback or suggestions.

Sincerely,

The authors

---

### Comment · Area_Chair_byTd · 2024-12-02
**Last day for reviewers' questions - Dec 2**

Dear reviewers,

This is the second and last reminder that December 2 is the last day for reviewers to ask questions to authors. As the paper received diverse ratings, can you confirm the authors' responses and post your updated thoughts by today?

Thank Reviewer rN5c for joining the discussion with the authors. As the authors replied for another round, could you check the authors' clarification and see whether they addressed your remaining concerns?

Thank you,

AC

---

### Meta-Review · Area_Chair_byTd · 2024-12-24

**Metareview:**

Summary: This paper introduced a benchmark to comprehensively evaluate the capabilities of MLLMs, which include multi-level sports reasoning tasks. Multiple baselines are considered for comprehensive evaluations.

Strengths: (1) The proposed benchmark could be helpful for further experiments for evaluation. (2) The evaluations are comprehensive. (3) The contribution to sports-related action understanding and reasoning area is clear.

The main remaining concerns (including the ones that reviewers did not reply to confirm with) are (1) the constructed dataset is not significantly different from previous methods, (2) the lack of diversity and coverage of the dataset, (3) implementation on reasoning prompt and rater selection could be possibly flawed.

This paper received diverse ratings, i.e., 6, 6, 5, 5. The AC checked the authors' responses, reviewers' comments, and the revised version. For the remaining concern (1), the authors claimed that "our dataset is the first to test MLLMs' sports understanding ability, particularly by combining action recognition with rule-based reasoning across different sports and varying difficulty levels", which AC agreed with. The novelty on sports-related QA with rule-based reasoning is clear and significant, which would be a good benchmark for sports-related action understanding and reasoning. For concern (2), the contributions to sports-related area are significant, and the goal is not a general benchmark. For concern (3), the authors provided results with reasoning prompt and rater selection. Also, for other concerns, the authors provided extensive results, which are comprehensive. Therefore, the AC think the major concerns have been addressed and thus recommend the paper as accept.

**Additional Comments On Reviewer Discussion:**

One reviewer pointed out that the constructed dataset is not significantly different from previous method. Please see matereview for details on this point.

---

### Decision · Program_Chairs · 2025-01-22

Accept (Poster)